# PLZF targets developmental enhancers for activation during osteogenic differentiation of human mesenchymal stem cells

Shuchi Agrawal Singh[1,2,3]*, Mads Lerdrup[1,3], Ana-Luisa R Gomes[1,3], Harmen JG van de Werken[4,5,6], Jens Vilstrup Johansen[1,3,7], Robin Andersson[1,3,7], Albin Sandelin[1,3,7], Kristian Helin[8,9,10], Klaus Hansen[1,3]*

[1]Biotech Research and Innovation Centre (BRIC), Faculty of Health and Medical Sciences, University of Copenhagen, Copenhagen, Denmark; [2]Department of Hematology, Cambridge Institute for Medical Research and Welcome Trust/MRC Stem Cell Institute, University of Cambridge, Cambridge, United Kingdom; [3]Centre for Epigenetics, Faculty of Health and Medical Sciences, University of Copenhagen, Copenhagen, Denmark; [4]Department of Cell Biology, University Medical Center, Rotterdam, Netherlands; [5]Cancer Computational Biology Center, University Medical Center, Rotterdam, Netherlands; [6]Department of Urology, University Medical Center, Rotterdam, Netherlands; [7]Department of Biology, The Bioinformatics Centre, University of Copenhagen, Copenhagen, Denmark; [8]The Novo Nordisk Center for Stem Cell Biology, Faculty of Health and Medical Sciences University of Copenhagen, Copenhagen, Denmark; [9]Cell Biology Program, Memorial Sloan Kettering Cancer Center, New York, United States; [10]Center for Epigenetics Research, Memorial Sloan Kettering Cancer Center, New York, United States

*For correspondence:
shuchiasingh@gmail.com;
sa796@cam.ac.uk (SAS);
klaus.hansen@bric.ku.dk (KHA)

Competing interests: The authors declare that no competing interests exist.

**Abstract** The PLZF transcription factor is essential for osteogenic differentiation of hMSCs; however, its regulation and molecular function during this process is not fully understood. Here, we revealed that the *ZBTB16* locus encoding PLZF, is repressed by Polycomb (PcG) and H3K27me3 in naive hMSCs. At the pre-osteoblast stage of differentiation, the locus lost PcG binding and H3K27me3, gained JMJD3 recruitment, and H3K27ac resulting in high expression of PLZF. Subsequently, PLZF was recruited to osteogenic enhancers, influencing H3K27 acetylation and expression of nearby genes important for osteogenic function. Furthermore, we identified a latent enhancer within the *ZBTB16/PLZF* locus itself that became active, gained PLZF, p300 and Mediator binding and looped to the promoter of the nicotinamide N-methyltransferase (*NNMT*) gene. The increased expression of NNMT correlated with a decline in SAM levels, which is dependent on PLZF and is required for osteogenic differentiation.

## Introduction

Human mesenchymal stem cells (hMSCs) possess self-renewal and multi-lineage differentiation potential toward osteogenic, chondrogenic and adipogenic specification (*Pittenger et al., 1999*; *Prockop, 1997*). Therefore, hMSCs represent a promising resource for regenerative medicine. However, for treatment efficiency and safety, it is instrumental to obtain a thorough understanding of basic molecular mechanisms that control the orchestrated activation of lineage-specific genes, determining cell fate of hMSCs and factors that ensure maintenance of the terminal differentiated state(s).

The adipogenic differentiation of MSCs have been extensively studied and have revealed that it consists of a cascade of transcriptional events that are driven by a series of transcription factors (*Rosen and Spiegelman, 2014*). However, the knowledge regarding osteogenic differentiation of MSCs is mainly limited to the activity of the key transcription factors RUNX2 and SP7 (osterix), and markers representing the different stages of differentiation such as proteins involved in matrix formation and mineralization (*Karsenty and Wagner, 2002*; *Neve et al., 2011*). Although it is well established that Polycomb (PcG)- and Trithorax (Trx)-group protein complexes are intimately linked to cell specification through gene repression and activation, respectively (*Piunti and Shilatifard, 2016*; *Schuettengruber et al., 2017*), very limited information regarding these complexes are available in naive hMSCs and cells undergoing osteogenic lineage specification. Moreover, there is a considerable lack in understanding of the chromatin changes associated with enhancer activation that leads to lineage-specific gene transcription during differentiation of mesenchymal stem cells.

Enhancers are cis-acting DNA regulatory elements that function as transcription factor (TF) binding platforms, increasing the transcriptional output of target genes through chromatin topological changes involving enhancer-promoter looping (*Calo and Wysocka, 2013*; *Plank and Dean, 2014*). Binding of lineage-specific TFs to enhancers is key to cell specification during stem cell differentiation and organism development (*Spitz and Furlong, 2012*). However, the enhancer landscape that becomes active and regulates lineage-specific gene expression during osteogenic commitment has not been fully explored.

The *ZBTB16* gene locus encodes a BTB/POZ domain and zinc finger containing TF known as PLZF (*Li et al., 1997*). Targeted deletion of *Zbtb16* in mice disrupts limb and axial skeleton patterning (*Barna et al., 2000*; *Fischer et al., 2008*), and although PLZF has been shown to be involved in osteogenic differentiation (*Djouad et al., 2014*; *Ikeda et al., 2005*), the underlying mechanisms of its function is only partly understood. PLZF has been described to have a dual function in transcription 1) as a gene repressor through interaction with HDAC1, mSin3a, SMRT and NCOR (*David et al., 1998*; *Hong et al., 1997*; *Melnick et al., 2002*; *Wong and Privalsky, 1998*), and PcG proteins (*Barna et al., 2002*; *Boukarabila et al., 2009*), 2) as a gene activator due to its positive impact on transcription (*Doulatov et al., 2009*; *Hobbs et al., 2010*; *Labbaye et al., 2002*; *Xu et al., 2009*). These studies have been performed in other tissues and cell types than MSCs, such as hematopoietic and germline cells. In these cell types, PLZF is already expressed at the stem cells stage and found to be required for the maintenance of the stem cell pool. However, in MSCs, PLZF is expressed only in differentiating MSCs and not in naive cells (stem cells), and its molecular function is so far unknown in these cells.

Through the use of genome-wide ChIP-sequencing and expression analysis, we now present evidence for a novel function of PLZF at developmental enhancers directing osteogenic differentiation of hMSCs. Interestingly, we find that the *ZBTB16* (PLZF) locus is bound and repressed by PcG proteins and extensively H3K27 tri-methylated (H3K27me3) in naive hMSCs. Upon osteoblast commitment of progenitor cells (pre-osteoblast stage), H3K27me3 demethylation takes place through JMJD3 (also known as KDM6B) recruitment to the *ZBTB16* locus, followed by extensive H3K27 acetylation and PLZF expression. Interestingly, upon expression, PLZF binds to a subset of enhancers, which gain H3K27 acetylation and correlates with induced expression of nearby genes important for osteogenic differentiation and function. Intriguingly, an initially PcG-repressed enhancer within the *ZBTB16* locus becomes active in pre-osteoblasts, gaining PLZF, p300 and Mediator binding. By use of Chromosome Conformation Capture combined with high-throughput sequencing (4C-seq) (*van de Werken et al., 2012b*), we show that this *ZBTB16* intragenic enhancer 'EnP' physically contacts the promoter of the nicotinamide n-methyltransferase gene (*NNMT*) and regulate its expression during osteogenic differentiation.

Taken together, our data reveal that the activation of the *ZBTB16* (PLZF) gene locus is an important event during osteogenic differentiation. PLZF regulates osteogenic differentiation of hMSCs through binding at gene enhancers and thereby it positively affects transcription of lineage-specific genes. The enhancer, EnP, localized within the *ZBTB16* locus is an example of a latent developmental enhancer that becomes active upon differentiation and regulates *NNMT* expression in a PLZF-dependent manner through Mediator and RNA-PolII recruitment and enhancer-promoter looping.

## Results

### The *ZBTB16* locus is activated during early osteogenic differentiation

To understand the transcriptional events that control the osteogenic differentiation of hMSCs, we performed RNA-seq analysis for global expression levels and ChIP-seq analysis for histone marks known to be associated with transcriptionally repressed (H3K27me3) and active (H3K4me3, H3K27ac) gene loci. We looked for the changes taking place at an early time point of differentiation (10 days) as compared to naive hMSCs (day 0). *Figure 1A* shows the timeline of osteogenic differentiation of hMSCs (*Kulterer et al., 2007*; *Neve et al., 2011*; *Qi et al., 2003*). The time point chosen for our analyses was based on prior microarray analyses (*Supplementary file 1*, *Figure 1—figure supplement 1B*), and an in situ calcium staining assay and western blots, which revealed day 10 as a timepoint where progenitor cells have committed to the osteoblast lineage, but cells are still far from the mineralization stage that takes place between day 15 and day 21 (*Figure 1—figure supplement 1A–C*). We will refer to the cells at this stage as immature osteoblasts (day 10) (*Figure 1A*). Two days after adding the osteogenic differentiation medium, the hMSCs are still proliferating (data not shown), which is supported by the expression of cyclin E and the presence of hyperphosphorylated pRB2 (p130) as revealed by western blotting (*Figure 1—figure supplement 1A*). Cells at this stage can be referred to as osteogenic committed progenitor cells (pre-osteoblasts) (*Figure 1A*) (*Neve et al., 2011*). While cells are still proliferating to some extent at day 5 (Cyclin E expression), they were completely arrested 10 days after induction of osteogenic differentiation (immature osteoblasts).

By analyzing the genome wide data revealing the most pronounced collective changes in histone marks and gene transcription, we observed a correlation between the loss of H3K27me3, a gain in H3K4me3 and increased gene expression at several genomic loci (*Figure 1—figure supplement 1D–H*, *Supplementary file 2*). The GO-terms for these groups of genes were associated to bone formation such as 'extracellular matrix', 'calcium' and 'osteogenesis' among others (*Figure 1—figure supplement 1I*). Examples of induced genes include 1) *BMP4, BMP6, IGF2, IGFBP2*; gene products known to promote osteogenic differentiation (*Hamidouche et al., 2010*; *Lavery et al., 2008*), 2) *Leptin, COMP, PPL*; genes encoding proteins involved in matrix formation (*Ishida et al., 2013*; *Turner et al., 2013*), 3) *ZBTB16, HIF3α*, examples of transcription factors involved in bone development (*Ikeda et al., 2005*; *Zhu et al., 2014*). Examples of ChIP-seq tracks and aligned RNA-seq data are shown in *Figure 1C* and *Figure 1—figure supplement 1J*. In contrast to differentiation of other cell types such as neuronal lineage, where a substantial number of gene loci showed dynamic changes in Polycomb repression (*Prezioso and Orlando, 2011*), H3K27me3 levels were surprisingly stable in differentiating hMSCs, and only few genes have significant changes in H3K27me3 levels at their TSS (*Figure 1B*, *Supplementary file 3*).

The most striking loss of H3K27me3 was found at the *ZBTB16* locus including intragenic exons and 3'UTRs (*Figure 1B and C*). Interestingly, the loss of H3K27me3 was accompanied by strong gain of H3K27ac across the whole *ZBTB16* locus and gain of H3K4me3 at the TSS, which was furthermore confirmed by ChIP-QPCR (*Figure 1C and D*).

### PLZF is expressed in osteoblast committed progenitor cells

The *ZBTB16* gene encodes the transcription factor PLZF, which has been shown to be critical for limb and axial skeleton patterning (*Barna et al., 2000*; *Fischer et al., 2008*). Intriguingly, our gene expression and histone modification data revealed that the *ZBTB16* gene locus was highly transcribed in response to osteogenic commitment of hMSCs, which is consistent with the loss of H3K27me3 in immature osteoblasts (day 10 of differentiation) (*Figure 2A,B*, and *Figure 1—figure supplement 2C*). Observing earlier timepoints, we found that the expression of *ZBTB16* was strongly induced ($\geq$ 4000 fold analyzed by QPCR) already at day 2 of differentiation which corresponds to pre-osteoblasts. We therefore analyzed the *ZBTB16* locus for the presence of PcG proteins and H3K27me3 levels at day 2 of osteogenic differentiation. Interestingly, we observed a loss of H3K27me3 and binding of the PRC2 subunit SUZ12 already in pre-osteoblasts (*Figure 1D and E* and data not shown). Furthermore, recruitment of the H3K27me3 demethylase JMJD3 was observed at the locus correlating to the loss of H3K27me3 (*Figure 1E*). These data revealed that derepression and activation of the *ZBTB16* locus and high expression of the PLZF (*Figure 2A and B*) was an early

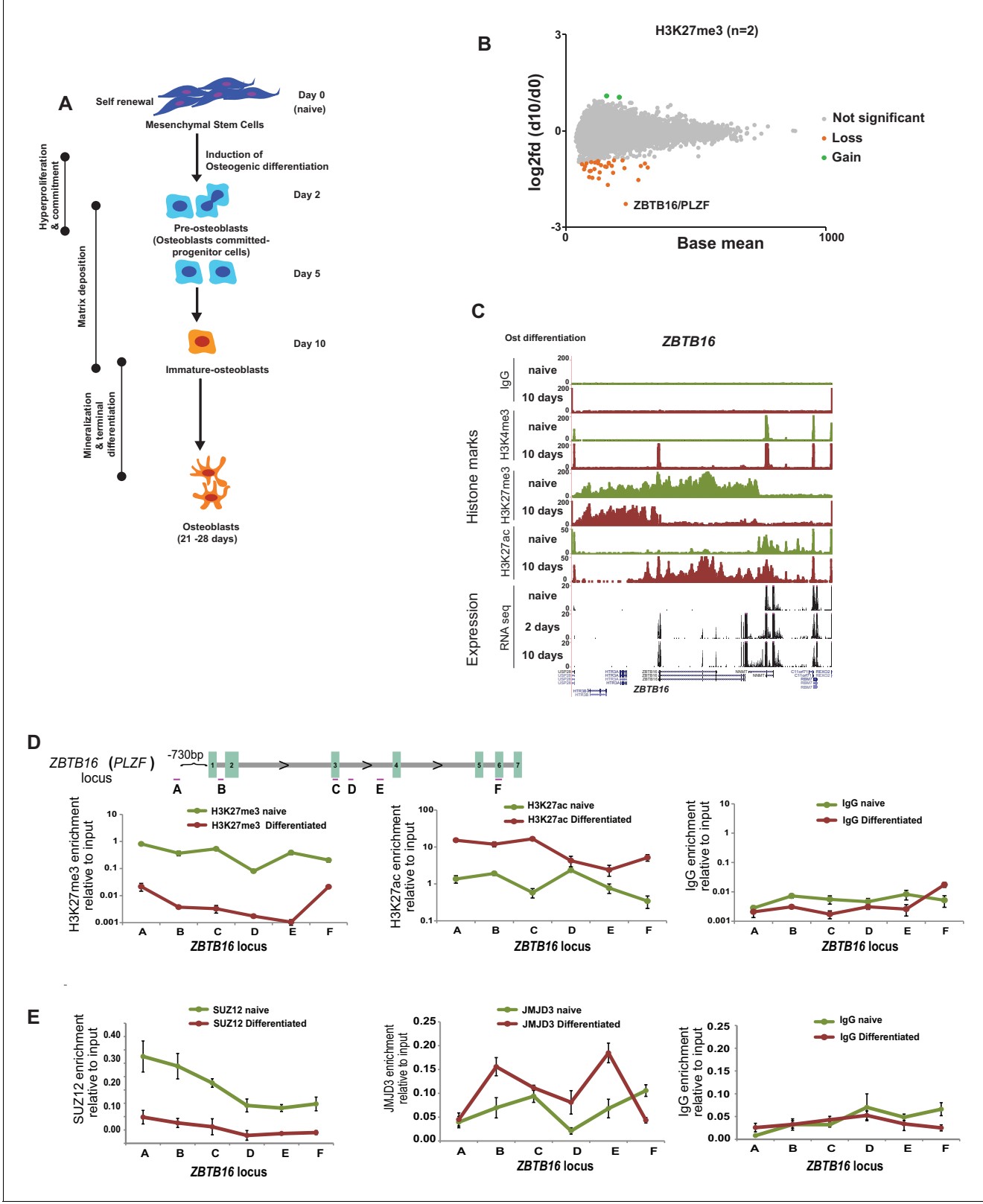

**Figure 1.** Epigenome and transcriptome profiling in hMSCs during osteogenic differentiation. (**A**) Schematic presentation of osteogenic (Ost) differentiation of hMSCs. (**B**) Scatter diagram of base mean (X-axis) and log2 fold differences in H3K27me3 levels (Y-axis) in immature osteoblasts relative to naive hMSC. Values were quantified from two replicates within ± 5 kbp of each TSS, and genes with a significant gain or loss were colored green or orange, respectively. ChIP-seq signal for H3K27me3 (biological replicates) in naive hMSCs and immature-osteoblasts (day 10 of osteogenic

*Figure 1 continued on next page*

*Figure 1 continued*

differentiation) were quantified at 8,822 TSS (±5 kbp), and analyzed by Deseq2. Green dots represent gene loci with gain of H3K27me3, while orange dots represent gene loci with loss of H3K27me3 (see details in Materials and methods and *Supplementary file 3*). As indicated, the largest loss of H3K27me3 was observed at the *ZBTB16/PLZF* locus in immature osteoblasts. (**C**) ChIP-seq tracks for histone marks and RNA-seq tracks for transcripts are shown for naive hMSCs (day 0) and immature osteoblasts for the *ZBTB16/PLZF* gene locus. (**D**) Upper part is a schematic presentation of the *ZBTB16* locus (an upstream region; exons 1–7; coding exons 2–7) showing the position of primers used for ChIP-QPCR (marked A-F). Lower panel represents ChIP-QPCR validation of H3K27me3 loss, and H3K27ac gain. The data shown represents three biological replicates shown as average of triplicate values from QPCR ± SD. (**E**) Loss of Polycomb (SUZ12) binding and increased JMJD3 enrichment after 2 days of osteogenic differentiation analyzed by ChIP-QPCR. The data shown represents three independent experiments and are averages of triplicate samples from QPCR ± SD. General IgG was used as control in ChIP-QPCR.

The online version of this article includes the following source data and figure supplement(s) for figure 1:

**Source data 1.** Validation of PLZF antibody in ChIP by PLZF kncokdown.
**Figure supplement 1.** After induction of differentiation, the hMSCs cells undergo a hyperproliferative phase within the first 2 days followed by growth arrest and differentiation.
**Figure supplement 2.** (A) The table shows the mean normalized expression (NE) values for indicated TF and markers obtained from RNA-seq during osteogenic differentiation of hMSCs based on three biological replicates.

event in osteogenic differentiation of hMSCs corresponding to the transition of stem cells to osteoblast committed progenitor cells (pre-osteoblasts).

Several markers that are expressed during osteogenic differentiation of hMSCs have been described. To understand how the induction of PLZF expression was timed relative to these, we investigated our RNA-seq data for known markers of osteogenesis. As summarized in *Figure 1—figure supplement 2A*, we observed that *RUNX2,* often referred to as the master regulator of bone formation (*Liu and Lee, 2013*; *Long, 2011*) was expressed already in proliferating naive hMSCs and that the expression was quite stable throughout the time course. Importantly, *SOX9* an essential factor for chondrocyte specification and known to antagonize RUNX2 transcriptional activity (*Loebel et al., 2015*; *Zhou et al., 2006*), was highly expressed in naive hMSCs, but was strongly reduced at day 2 of osteogenic differentiation. This down regulation of *SOX9* likely reflects the transition to osteoblast committed progenitor cells. Furthermore, *FOXO1*, previously shown to be involved in osteogenic differentiation (*Teixeira et al., 2010*) was induced at day 2. Importantly, transcription factors of other lineage fates such as *MYOD1* (myocytic commitment) and *SOX5/8* (chondrocytic commitment)(*Chimal-Monroy et al., 2003*) were repressed and marked by H3K27me3 in naive hMSCs and still at day 10 of osteogenic differentiation (*Figure 1—figure supplement 2A*). In addition, factors known to mark the transition from the pre-osteoblast stage to immature osteoblasts such as *ALPL* (alkaline phosphatase) and *SPP1* (osteopontin) (*Aubin et al., 1995*; *Huang et al., 2007*) were strongly induced at day 10 of osteogenic differentiation (*Figure 1—figure supplement 2A,B*), while *SP7* (osterix) and *BGLAP* (osteocalcin), markers of the mineralization stage (*Aubin et al., 1995*; *Harada and Rodan, 2003*) were still repressed and marked by H3K27me3 in immature osteoblasts (*Figure 1—figure supplement 2A,B,C* and data not shown). In conclusion, our analyses position PLZF expression as a very early event during osteogenic differentiation and likely implicated in the transition from naive hMSCs to osteoblast committed progenitor cells (pre-osteoblasts).

## PLZF is recruited to distal regulatory regions and affects expression of osteogenic-specific genes during differentiation of hMSCs

To understand the function of PLZF in osteogenic differentiation of hMSCs, we performed ChIP-seq against PLZF in naive proliferating (day 0) and immature osteoblasts (day 10 of differentiation). We identified 2,282 PLZF peaks that were observed only in immature osteoblasts (*Figure 2C*, *Supplementary file 4*). The analysis revealed that the majority of PLZF-binding sites localized at intergenic (45.5%) and intronic regions (34.7%) combined representing 1,830 out of 2,282 PLZF-binding sites (*Figure 2D*). Very few PLZF-binding sites were observed at gene promoters (± 1 kb of TSS: 3.0%) and upstream (−10 kb from TSS: 8.5%) or downstream (+ 10 kb from TSE: 4.8%) regions (*Figure 2D*). Discriminative *de novo* motif analysis of the DNA sequences within the PLZF peaks, compared to sequences from negative control regions, revealed a significant occurrence of a TACAGC motif (E = $1.1 \times 10^{-81}$), which is highly similar to the TAC(T/A)GTA PLZF motif identified

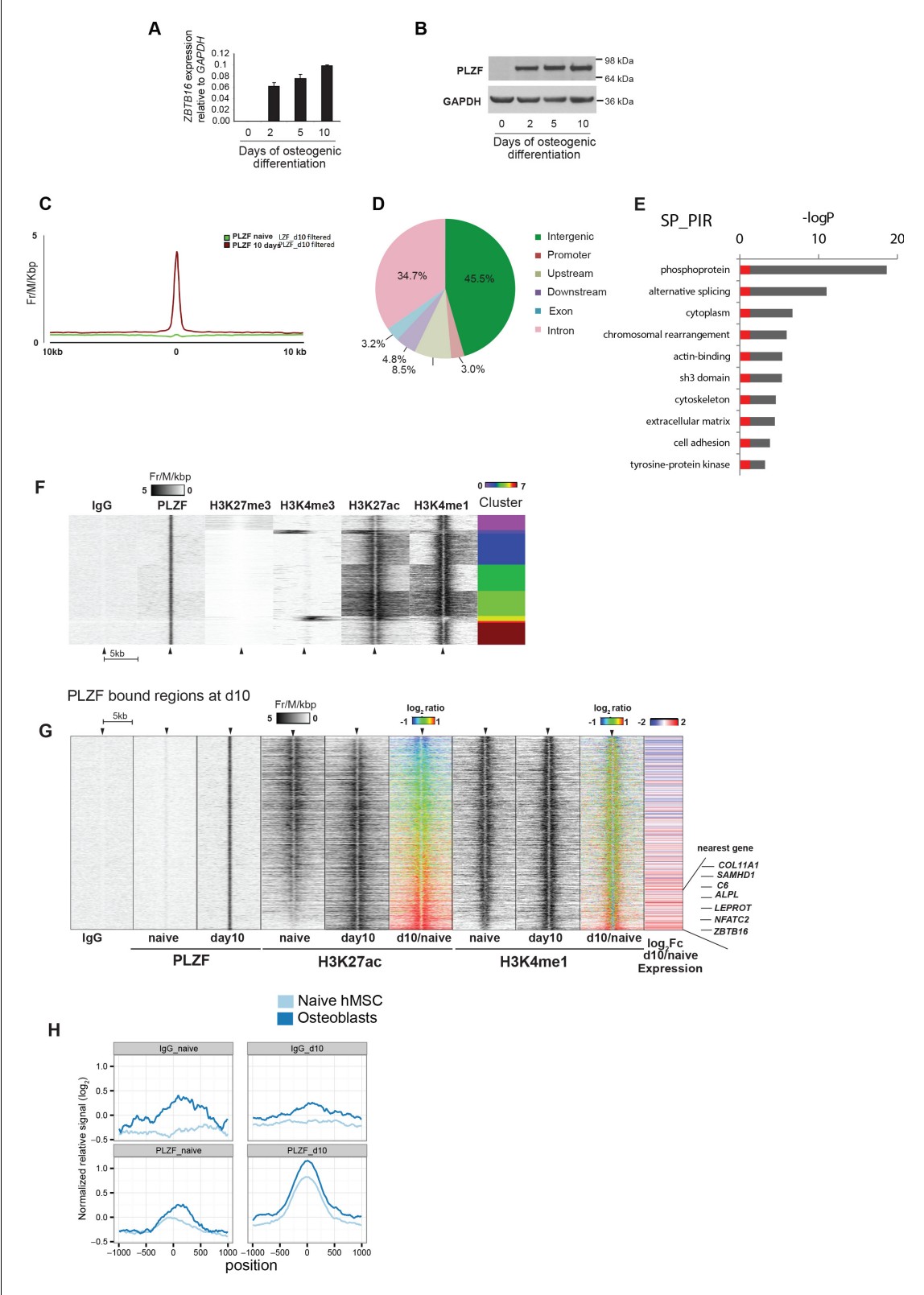

**Figure 2.** PLZF induction and binding to enhancers during osteoblasts commitment of progenitor cells. (A) PLZF expression during early osteogenic differentiation of hMSCs shown by (A) RT-QPCR (*ZBTB16*) and (B) western blot. GAPDH was used as a loading control. (C) Average tag density plot for PLZF peaks from ChIP-seq in naive hMSCs or immature-osteoblasts (day 10). Peak finding revealed 2,282 specific PLZF peaks in immature-osteoblasts (*Supplementary file 4*). See Supplementary Methods for details regarding peak finding analyses. (D) Pie chart showing the percentage of PLZF peaks

*Figure 2 continued on next page*

*Figure 2 continued*

(n = 2,282) that overlapped with different genomic regions; promoter (−1 kb to TSS, n = 70); upstream (−10 kb to TSS excluding promoters, n = 195); downstream (+10 kb from the end of the gene, n = 110); exon (n = 75); intron (n = 792); and intergenic (> ± 10 kb from TSS or end of the gene, n = 1,040). (E) GO term analyses of genes nearby to PLZF peaks (n = 2,282) observed by DAVID (*The Database for Annotation, Visualization and Integrated Discovery*) (Huang da, *Bailey et al., 2009*). The key words enriched are displayed in the plot. The Y-axis represents the Benjamini Hochberg corrected –LogP value. Grey parts of the bars are above a 0.05 cut-off. (F) Heat-maps at PLZF peaks (n = 2,282) clustered according to local densities of the H3K27me3, H3K4me3, H3K27ac and H3K4me1 histone marks in immature-osteoblasts. ChIP-seq signals from each antibody were normalized to library sizes and visualized at PLZF peaks ± 5 kb. Only PLZF peaks specific for day 10 of osteogenic differentiation were included, and peaks with PLZF signal at day 0 were excluded (for threshold and genomic peak positions please see *Supplementary file 3*, and Supplementary methods). (G) Heat maps showing the changes in H3K27ac and H3K4me1 at PLZF peaks (n = 2,282), and expression of the nearest gene in naive hMSCs (day 0) and immature osteoblasts. The distribution of normalized H3K27ac or H3K4me1 ChIP seq signal in naïve hMSCs or immature osteoblasts is shown in the heat maps, centered at PLZF peaks identified in immature osteoblasts (± 5 kb). The ratiometric heatmap depicts the changes in H3K27ac and H3K4me1 levels in immature-osteoblasts compared to naive hMSCs ($\log_2$fc day10/naïve) and all heatmaps were sorted according to the H3K27ac ratio with the highest day10/naive H3K27ac ratio at the bottom. Gain is colored red ($\log_2$ fold ≥ 1, corresponds to 403 PLZF peaks), no change is colored green ($\log_2$ fold>-1 and < 1, n = 1,844 PLZF peaks) and loss is colored blue ($\log_2$ fold ≤ 1, n = 35 PLZF peaks). The RNA-seq based heat-map shows the $\log_2$fc in expression (day10/naive) of nearest transcript mapped to the center of individual PLZF peaks. Transcripts with induced expression are colored red, while blue coloring reflects decreased expression between naïve hMSCs and immature osteoblasts. Examples of genes with increased expression correlating with increased H3K27ac and PLZF binding at nearest genomic region (enhancer) are indicated on the right side. *COL11A1*, Collagen11A1; *SAMHD1*, SAM domain and HD domain1; C6, Complement Component 6; NFATC2, nuclear factor of activated T-cells; *ALPL*, Alkaline Phosphatase; *LEPROT, Leptin Receptor Overlapping Transcript; ZBTB16*, Zinc Finger and BTB Domain Containing Protein 16. (H) Average normalized ChIP-seq pile-up signal from PLZF or IgG control in naive hMSCs or immature-osteoblasts,± 1 kb around midpoints of enhancers (defined by FANTOM5) transcribed in mesenchymal stem cells (naive hMSCs) and osteoblasts. Pile-up signal was normalized to the average base pair signal across all FANTOM5 enhancers (n = 43,011).

The online version of this article includes the following source data and figure supplement(s) for figure 2:

**Figure supplement 1.** (A) Top motifs and output from discriminative de novo motif analysis of d10 PLZF peaks and random negative control regions matched to the peaks in terms of TSS distance and orientation.

**Figure supplement 1—source data 1.** Validation of PLZF antibody in ChIP by PLZF kncokdown.

**Figure supplement 2.** (A) Average tag distribution plots show co-occurrence of PLZF with Polycomb (SUZ12, RING1B) or H3K27me3 in hMSCs before and after osteogenic differentiation for 10 days (immature-osteoblasts) centered at PLZF peak ± 5 kb.

---

by Ivins et al. (*Figure 2—figure supplement 1A and B*) (*Ivins et al., 2003*). Gene ontology analyses of closest genes to PLZF bound regions enriched for terms highly relevant for osteogenic differentiation such as 'cytoskeleton' and 'extracellular matrix' among others (*Figure 2E*). The specificity of the PLZF antibody in ChIP was confirmed by PLZF knockdown in hMSCs followed by induction of differentiation and ChIP-QPCR at selected loci (*Figure 2—figure supplement 1C–E*).

## PLZF binds to active chromatin regions in immature osteoblasts

To study the relationship of PLZF binding at genomic sites and histone modifications, we performed a cluster analysis relating PLZF-binding sites to ChIP-seq data for H3K27me3, H3K4me3, H3K4me1 and H3K27ac histone marks in immature osteoblasts. It has previously been suggested that PLZF functions as a repressor of gene transcription and has a link to PcG recruitment. Therefore, we were surprised to observe a very limited co-occurrence of PLZF with PRC1 (RNF2) and PRC2 (SUZ12) as well as the H3K27me3 repressive mark catalyzed by PRC2 (*Figure 2F* and *Figure 2—figure supplement 2A*). In contrast, there was a clear co-occurrence of PLZF binding, with H3K27ac and H3K4me1, histone marks characterizing active gene enhancers (H3K27ac, H3K4me1) (*Creyghton et al., 2010*) and promoters (H3K27ac). However, the number of gene promoters bound by PLZF was very low (*Figure 2D*), indicating that the majority of PLZF-bound regions represent active gene enhancers.

## PLZF is recruited to gene enhancers in immature osteoblasts

Activating transcription factors often recruit histone acetyl transferases (HATs) leading to H3K27 acetylation as part of the gene activation mechanism (*Spitz and Furlong, 2012*). To investigate if PLZF during osteogenic differentiation would lead to increased H3K27ac at its binding sites, we next analyzed the levels of H3K27ac before and after PLZF recruitment genome wide. As evident in the ratio-metric heat map (*Figure 2G*), approximately 18% of the 2,282 genomic sites binding PLZF, gained H3K27ac (≥ 2 fold, *Supplementary file 5*) upon osteogenic differentiation (red-colored

population), whereas 81% of PLZF-bound regions retained their H3K27ac pattern (green-colored population). A small subset of PLZF-bound regions (approximately 1%) showed reduction in H3K27ac (($\geq$ 2 fold, blue-colored population). In contrast, the H3K4me1 mark was relatively constant at the PLZF-bound regions for the majority of sites (n = 2,123; 93%) when comparing immature osteoblasts to naïve hMSCs (*Figure 2G*). Interestingly, when comparing the PLZF bound enhancers that gained H3K27ac to those that retained H3K27ac levels and related it to the expression of nearest transcript, we observed a strong correlation between gain of H3K27ac and increased expression of proximal genes. Gene Ontology analyses revealed that many of these genes were related to 'bone development', 'osteoclast differentiation' (rightmost panel in *Figure 2G* and examples of ChIP-seq tracks in *Figure 2—figure supplement 2B,C* and *Supplementary file 5*). This suggested that a subset of PLZF-bound enhancers become active for the regulation of nearest gene that relates to an increase in H3K27ac. Furthermore, some of the PLZF-binding sites with unchanged levels of H3K27ac, showed increased expression of nearest gene upon induction of osteogenic differentiation. This might be related to other chromatin features, for example, other histone modifications and/or cooperative TF-binding effects at individual enhancers (*Figure 2G*) (*Spitz and Furlong, 2012*).

## PLZF localizes to eRNA expressing active enhancers

Active enhancers produce so-called enhancer RNAs (eRNAs), which can be detected by global 5' end RNA sequencing (CAGE). In the FANTOM5 project, this technique was used to create an atlas of 43,011 CAGE-predicted active enhancer regions across numerous cell types (432 primary cell types and 241 cell lines) and tissues (135 tissues) (*Andersson et al., 2014*). When comparing the PLZF-binding sites in immature osteoblasts with the CAGE-derived enhancer candidates, we observed that 603 out of 2,282 PLZF-binding sites (26%) localized within ± 1 kb of a predicted FAN-TOM5 enhancer, corresponding to 740 enhancers (*Supplementary file 6*). Interestingly, PLZF-bound enhancers in our hMSCs ChIP-seq data were highly enriched for enhancers that were significantly expressed in FANTOM CAGE libraries, of naive hMSCs (p < 5*10$^{-8}$, Fisher's exact test) and osteoblasts (p < 4*10$^{-39}$, Fisher's exact test) (*Figure 2H*) (it should be noted that the time point 10 days of osteogenic differentiation, immature osteoblasts, does not exist in the FANTOM5 data). Interestingly, almost equal numbers of PLZF-bound enhancers identified in our ChIP-seq analysis in immature osteoblasts were CAGE positive (FANTOM5 data) in naive hMSCs or in osteoblasts (13.2%: 98 out of 740% and 13.6%: 101 out of 740, respectively) and approximately 42% (42 out of 98 or 101) of these enhancers are in common between naive hMSCs and osteoblasts. This suggests that 1) in naive hMSCs, where PLZF is absent, other TFs define a subpopulation of active enhancers that later on during osteogenic differentiation is bound by PLZF and 2) that PLZF likely continues to be bound to a significant number of such enhancers in terminal differentiated osteoblasts. Moreover, the data suggest that there exist a number of lineage-specific enhancers that become active and gain PLZF binding in the process of differentiation (*Figure 2—figure supplement 2B*).

## PLZF affects histone H3K27ac at bound enhancers and an osteogenic gene expression signature

To further investigate the relationship between PLZF recruitment to enhancers and changes in H3K27ac genome wide, we performed siRNA-mediated PLZF knockdown in naïve hMSCs followed by induction of osteogenic differentiation, and H3K27ac ChIP-seq. We tested three different siRNA oligos targeting PLZF (*Figure 3—figure supplement 1*). All three oligos showed significant knockdown of PLZF mRNA compared to non-targeting control siRNA. We chose oligo #57 for further experiments. Since it is not possible to maintain a stable knockdown of PLZF for 10 days of differentiation using siRNA, we analyzed the changes in H3K27ac at day 2 of differentiation (pre-osteoblasts), where PLZF knockdown was still pronounced (*Figure 3A*). Clustering of the data revealed a highly reproducible increase in H3K27ac in pre-osteoblasts, which was reduced when PLZF expression was down-regulated by siRNA (*Figure 3B*).

Examples of regions showing a gain of H3K27ac in a PLZF-dependent manner are shown as ChIP-seq tracks in *Figure 3C*. In addition to regions that gained H3K27ac, there was furthermore a fraction of regions that lost H3K27ac upon osteogenic differentiation (blue in left panel). However, only

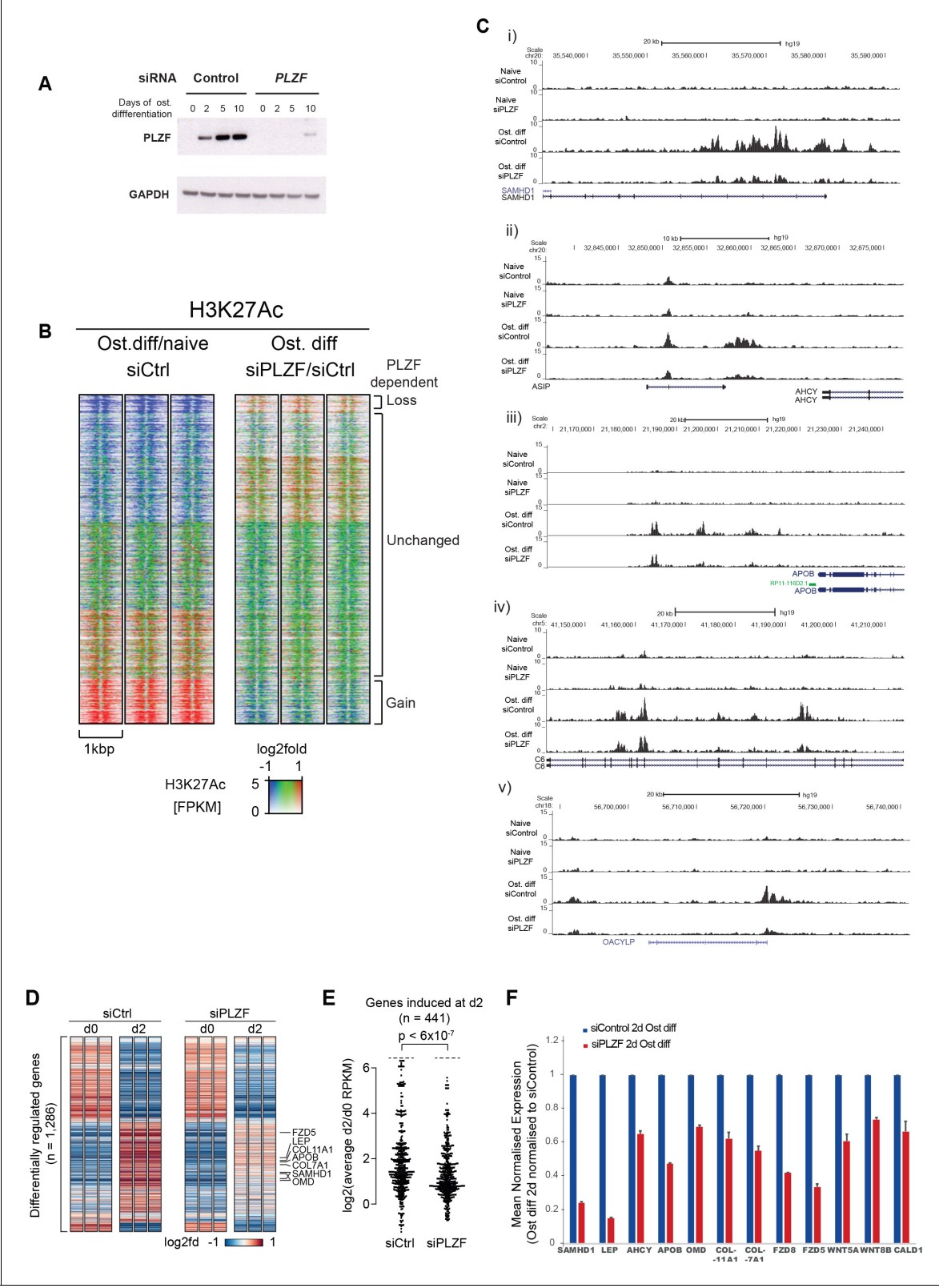

**Figure 3.** PLZF-dependent H3K27 acetylation and osteogenic-specific gene expression. (**A**) Time course of PLZF expression before and after siRNA mediated knockdown in naïve and osteogenic-induced hMSCs by western blot. GAPDH was used as loading control. (**B**) Ratiometric heat maps showing the changes in H3K27ac at PLZF peaks (n = 2,282) in control (siCtrl, left side) or the effect of PLZF knockdown (siPLZF, right side) in hMSCs. The distribution of normalized H3K27ac ChIP seq signal in naive or in 2-day osteogenic differentiation-induced hMSCs is shown, centered at PLZF peaks

*Figure 3 continued on next page*

*Figure 3 continued*

(± 500 bp). Regions were clustered according to the log2fd (compared to average values for all conditions) in each of the three biological replicates for each condition. The regions that loose H3K27ac upon induction of osteogenic differentiation are colored in blue while regions that gain H3K27ac are marked red. Regions without changes are depicted in green. (C) Genome browser tracks representing examples of genomic regions that gain H3K27ac in a PLZF-dependent manner upon induction of osteogenic differentiation (day 2). (D) Genes induced in osteoblast commited progenitor cells had reduced expression in the absence of PLZF, as observed by RNA-seq. Heatmaps representing the RPKM-values of differentially regulated genes from RNA-seq of PLZF knock down or control siRNA transfected hMSCs (done in three biological replicates) at 2 days of osteogenic induction. RPKM-values are log2 normalized to the average signal of all 12 samples (Control siRNA and PLZF siRNA transfected from naive and osteogenic induced 2d). The vertical order is similar to the clustered heatmap from differentially regulated genes from microarray analyses shown in *Figure 1—figure supplement 1B* (n = 1286 genes). (E) Beeswarm plots comparing the log2 fold difference between PLZF knock down or control siRNA transfected hMSCs harvested at day 0 and day 2. Plots shows the average RPKM values from RNA-seq performed in three biological replicates, and the p-value is calculated using a Mann-Whitney U-test. (F) The mean normalized expression of selected osteogenic lineage-specific genes in hMSCs after induction of osteogenic differentiation (2 days) in control siRNA or PLZF siRNA-transfected hMSCs. The values are averages from three biological replicates ± SD. SAMHD1, SAM and HD domain containing deoxynucleoside triphosphate triphosphohydrolase 1; LEP, Leptin; AHCY, Adenosylhomocysteinase; OMD, Osteomodulin; COL11A1, Collagen Type XI Alpha 1 Chain; COL7A1, Collagen Type VII Alpha 1 Chain; FZD8, Frizzled Class Receptor 8; FZD5, Frizzled Class Receptor 5; WNT5A, Wingless-Type MMTV Integration Site Family, Member 5A; WNT8B, Wingless-Type MMTV Integration Site Family, Member 8B; CALD1, Caldesmon 1.

The online version of this article includes the following source data and figure supplement(s) for figure 3:

**Source data 1.** Mean normalised expression of selected transcripts from RNA seq analyses.

**Figure supplement 1.** (A) PLZF knockdown using three different siRNA (55, 56, 57) in hMSCs measured by RT-QPCR.

---

a small fraction of these regions showed PLZF-dependent changes that are in line with the previous reported function of PLZF as a transcriptional repressor (*Supplementary file 7*).

To further gain insight to the transcriptional changes taking place in the absence of PLZF during osteogenic differentiation, we performed a whole transcriptome analysis by RNA-seq after siRNA-mediated PLZF knockdown in hMSCs. There were 1,897 genes differentially regulated upon osteogenic induction in control siRNA-transfected hMSCs compared to 1,184 genes differentially regulated in PLZF knockdown cells. Only 44% (950/1,897) of these genes overlapped between control siRNA and PLZF knockdown cells (*Figure 3—figure supplement 1B*). As shown in the *Figure 3D and E*, the PLZF knockdown significantly reduced the expression of genes that were normally induced in osteoblast committed progenitor cells (*Figure 3D and E*, *Supplementary file 8*). The heatmaps in *Figure 3D*, representing the RPKM-values of differentially regulated genes from RNA-seq of PLZF knockdown or control siRNA (in biological triplicates) in hMSCs. The vertical order of genes is similar to the clustered heatmap shown in *Figure 1—figure supplement 1B*, (differentially regulated genes from microarray analyses). The RNA-seq data clearly indicate; i) a high degree of reproducibility in biological replicates, ii) pronounced impact of PLZF knockdown on gene regulation at early stage (2 days) of osteogenic differentiation. The expression of upregulated genes is mainly affected by the PLZF knockdown. Furthermore, in *Figure 3E*, the plot shows that PLZF knockdown significantly affected the osteogenic transcriptional program (p < 0.0001) when comparing the log2 fold differences between PLZF knockdown or negative control siRNA in hMSCs, harvested at day 0 and day 2. The Gene Ontology (GO) analyses of differentially regulated genes in control siRNA transfected cells enriched the terms such as 'extracellular matrix organization, regulation of cell differentiation and ossification'. Whereas genes regulated in PLZF knockdown hMSCs, enriched terms including 'Interferon signaling and chondrocyte differentiation' (*Figure 3—figure supplement 1C*). Moreover, we looked for the influence of PLZF knockdown on expression of genes proximal to the PLZF-bound enhancers that gain H3K27ac. Interestingly, we observed that lack of PLZF in many cases correlated with the reduced expression of nearby genes to the regions that gain H3K27ac in PLZF-dependent manner (examples are shown in *Figure 3C and F*).

## The PLZF-bound developmental enhancer EnP becomes active upon osteogenic differentiation of hMSC

Interestingly, among the PLZF-bound genomic regions, we identified the *ZBTB16* locus itself (*Figure 4A*). Multiple PLZF peaks appeared across the intronic region covering around 50 kb, and flanking exons 3 and 4. We also observed a remarkable gain of H3K27ac at these regions overlapping with PLZF peaks in immature osteoblasts. In contrast, H3K27ac was absent in naive hMSCs and

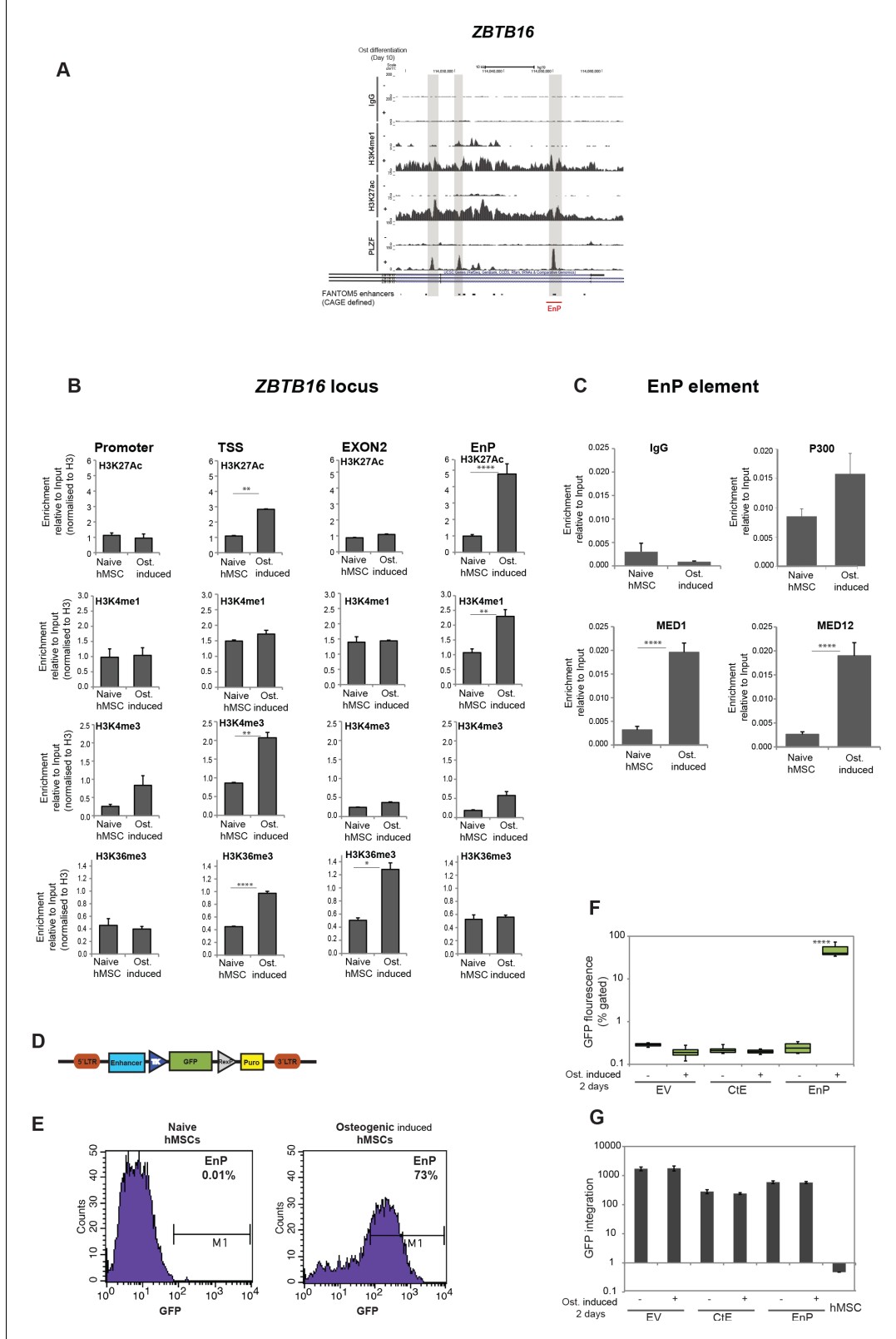

**Figure 4.** Characterization of a PLZF-bound, *ZBTB16* intragenic enhancer. (**A**) Genome browser tracks representing the genomic region within the *ZBTB16* locus that gain H3K4me1, H3K27ac and PLZF binding in immature osteoblasts (highlighted in grey box). The overlap with FANTOM5 enhancers is indicated at the bottom of the tracks. (**B**) ChIP-QPCR at the *ZBTB16* locus showing enrichment of H3K27ac, H3K4me1, H3K4me3 and H3K36me3 histone marks in naive hMSCs and in pre-osteoblasts (2 days of osteogenic differentiation). Data showing mean ± SD from triplicates of QPCR from

*Figure 4 continued on next page*

*Figure 4 continued*

three biological replicates, ****p < 0.0001, **p = 0.0025 for H3K27ac, not significant for other regions; **p = 0.0092 for H3K4me1, not significant for other regions; **p = 0.0071 for H3K4me3 at TSS, not significant for other regions; ****p < 0.0001, *p = 0.027 for H3K36me3, not significant for other regions; calculated by two-way ANOVA with Sidak's multiple comparison tests. The regions (promoter, TSS, EnP) amplified using primers A, B and E shown in *Figure 1E* and primers binding at Exon 2 of the *ZBTB16* locus. (**C**) ChIP-QPCR for the Mediator components MED1 and MED12 as well as p300 in naive or in pre-osteoblasts (2 days of osteogenic differentiation) at the EnP element. Data shows mean ± SD from triplicates of QPCR from three biological replicates, **p < 0.001 calculated by multiple t-tests with FDR 1% and two-stage step-up method of Benjamini, Krieger and Yekutieli. (**D**) Schematic presentation of the lentiviral enhancer GFP-reporter system used (pSINMIN). EnP (region corresponding to the largest PLZF peak in *Figure 4A*) or a control region (CtE; region within *ZBTB16* locus distal to EnP but without a PLZF peak) was cloned upstream of the minimal TK promoter driving the expression of GFP. (**E**) Histogram represents GFP expression measured by flow cytometer in hMSCs cells transduced with lentivirus encoding EnP-GFP followed by induction of osteogenic differentiation (2 days). Data is a representative of five biological replicates. The percentage of GFP positive cells from gated live cells are indicated in the upper right corner. (**F**) Box plot represents the GFP expression obtained by flow cytometer measurements in hMSCs transduced with empty vector (EV), control element (CtE) or enhancer (EnP) cloned in the pSINMIN GFP reporter, before and after induction of osteogenic differentiation for two days. The data shows median calculated from five biological replicates. Median is shown by horizontal line. ****p < 0.0001 calculated by 2-way ANOVA with Sidak's multiple comparison tests. (**G**) Integration of the GFP coding sequence was analyzed by QPCR using primers for GFP on genomic DNA isolated from each group of samples. Data shown are mean ± SD from three biological replicates. Untransduced hMSCs were used as a negative control and shown as right most bars in the plot.

The online version of this article includes the following source data and figure supplement(s) for figure 4:

**Source data 1.** ChIP for Med1, Med12 and P300 in hMSCs.
**Source data 2.** H3K4me3 ChIP in hMSCs.
**Source data 3.** H3K4me1 ChIP in hMSCs.
**Source data 4.** H3K27ac ChIP in hMSCs.
**Source data 5.** H3K36me3 ChIP in hMSCs.
**Source data 6.** Med12 ChIP in hMSCs.
**Source data 7.** Med1 ChIP in hMSCs.
**Source data 8.** GFP reproter integration analyses.
**Figure supplement 1.** (A) ChIP-QPCR at another PLZF peak observed within the ZBTB16 locus showing enrichment of histone marks (H3K27ac, H3K4me1, H3K4me3 and H3K36me3) in naive hMSCs or in pre-osteoblasts (at day 2 of osteogenic differentiation).

the region was heavily marked by H3K27me3. Intriguingly, two out of three observed peaks over-lapped with CAGE defined enhancers. We refer to the enhancer with the most significant PLZF peak as 'EnP' (Enhancer within PLZF locus bound by PLZF) and decided to further characterize it as an example of a PLZF-bound osteogenic enhancer. To validate, that the PLZF-bound intragenic EnP element correspond to a functional enhancer, we examined the presence of histone marks and gene regulatory factors known to bind enhancers by ChIP QPCR. First, we analyzed a number of relevant histone modifications and confirmed that the EnP element gained H3K4me1 and H3K27ac in pre-osteoblasts as compared to naive hMSCs, but contained low levels of H3K4me3 and H3K36me3 compared to other regions within the *ZBTB16* locus (*Figure 4B*). The gain in H3K4me3 and H3K27ac was observed at the TSS, and H3K36me3 was more enriched at exon two within the *ZBTB16* locus in pre-osteoblasts correlating with transcription of the gene (*Figure 4B*). We obtained similar results when investigating another PLZF peak within the *ZBTB16* locus (*Figure 4—figure supplement 1A*). It is well established that binding of the CBP/p300 histone acetyl transferases (HATs) marks enhancers and catalyzes H3K27 acetylation (*Visel et al., 2009*). Moreover, the binding of lineage-specific TFs together with the Mediator co-activator complex at enhancers facilitates RNA polymer-ase II (RNA PolII) recruitment to the promoter of target genes (*Carlsten et al., 2013*). We therefore, performed ChIP for p300 and the Mediator co-activator complex (MED1, MED12) in naive hMSCs and in pre-osteoblasts. The data revealed enrichment of all three factors at EnP (*Figure 4C*). Taken together, our data strongly suggest that the EnP element that lies within the *ZBTB16* locus repre-sents a latent (no H3K4me1 and no H327ac) PcG repressed enhancer in naive hMSCs which gains the characteristics of an active enhancer in pre-osteoblasts.

To further validate EnP as a differentiation-induced enhancer, we analyzed the enhancer activity of EnP in a GFP reporter assay (*Figure 4D*). As a control element of similar size, we cloned another upstream distal region from the *ZBTB16* locus (CtE) that did not show PLZF binding. There was a complete absence of GFP fluorescence in hMSCs infected with the CtE-GFP element or empty vec-tor control (*Figure 4E and F*). Interestingly, EnP showed activity in a differentiation-specific manner, being completely inactive in naive hMSCs (*Figure 4E and F*). These results demonstrated that the

EnP element likely represents a latent repressed enhancer in naive hMSCs, which upon induction of osteogenic differentiation acquires the properties of an active enhancer and can drive the expression of a nearby gene. To ensure that the integration of the lentiviral reporter constructs was comparable for EnP-GFP and CtE-GFP, we performed a QPCR for the GFP coding part on genomic DNA (gDNA) isolated from transduced cells. The data confirmed that the integration efficiency of the two reporter constructs were highly comparable and therefore the increase in GFP fluorescence in the EnP-GFP infected hMSCs was due to the functional activity of the EnP enhancer in response to induction of osteogenic differentiation (*Figure 4G*).

## The PLZF-bound intragenic enhancer (EnP) loops to the promoter of the neighboring nicotinamide N-methyl transferase gene (*NNMT*) gene

It is widely accepted that enhancers interact with gene promoters through chromatin looping and thereby regulate the transcriptional activity of associated genes (*de Laat and Duboule, 2013*; *Levine et al., 2014*). Therefore, we decided to use 4C-seq in order to obtain an unbiased genome-wide screen for DNA contacts made by the EnP element upon induction of osteogenic differentiation. The experiment was performed using naïve and pre-osteoblasts in biological replicates. Interestingly, the 4C-seq data revealed several contacts between the EnP element and other genomic sites observed within a 1 Mb window. We found that the EnP element looped to the promoter of the *NNMT* gene, which is located ~100 kb downstream of the EnP element (*Figure 5A*). Furthermore, this contact was specific for pre-osteoblasts, since no looping was detected in naïve hMSCs (*Figure 5A*). This is in agreement with the *ZBTB16* locus being repressed by PcG and H3K27me3 in naive hMSCs and the inaccessibility of the EnP enhancer. To test the reliability of our 4C-seq data, we repeated the experiment under similar conditions, but using the *NNMT* promoter as viewpoint for the analysis. The results confirmed the contact of the *NNMT* promoter with the EnP element within the *ZBTB16* locus and that it was specific for hMSCs undergoing osteogenic differentiation (*Figure 5B*). Interestingly, a rather extended contact area around EnP (> 10 kb) was observed when the *NNMT* promoter was used as viewpoint. Additionally, a *NNMT* upstream region and the promoter of *ZBTB16* were used as a negative 4C-seq controls. The *NNMT* upstream region showed limited, but much less than *NNMT*-promoter enriched contact frequencies with the EnP element, whereas the *ZBTB16* promoter showed no enrichment of contact frequencies interaction with EnP. Thus this may represent a Polycomb repressed compacted region, since the whole locus was covered by H3K27me3 (*Figure 5—figure supplement 1A and B*). In conclusion, these data support the existence of what we believe could be a cluster of enhancers localized within the *ZBTB16* gene locus, where we observed a strong and broad H3K27ac enrichment in ChIP-seq and multiple PLZF peaks (intron 2–4 in *Figure 4A*). This suggests, that beside EnP several other intragenic elements within the *ZBTB16* locus have the potential to act as gene enhancers.

## The *ZBTB16* intragenic enhancer EnP regulates the *NNMT* gene promoter

In order to reveal whether the contact of the EnP enhancer with the *NNMT* promoter had an impact on *NNMT* expression, we measured mRNA and protein levels. Indeed, *NNMT* expression was increased upon induction of osteogenic differentiation of hMSCs (*Figure 6A and B*). Combined with the 4C-seq analyses these data suggested a positive influence of the distal EnP enhancer located within the *ZBTB16* locus on *NNMT* expression.

The *NNMT* gene encodes the nicotinamide N-methyl transferase, a metabolic enzyme that regulates SAM (S-adenosyl-methionine) homeostasis. NNMT functions in a biochemical reaction where it transfers the methyl group from SAM to nicotinamide and converts SAM to SAH (S-adenosyl-homocysteine) (*Figure 6C*) (*Aksoy et al., 1995*). To investigate whether the increase in *NNMT* expression upon osteogenic differentiation affected the overall SAM levels in the cells, we measured SAM using a fluorescence-based assay. Interestingly, we observed a decrease in SAM levels upon differentiation, suggesting a correlation between increase in *NNMT* expression and a decrease in SAM levels (t test, p = 0.01, *Figure 6D*). However, the drop in SAM levels was only minor, although significant (p = 0.01 by t-test), suggesting a delicate balance on SAM homeostasis during osteogenic differentiation. SAM being a critical co-factor for methylation of DNA through the action of DNA methyl transferases (DNMTs) and histones through the activity of histone methyl transferases (HMTs)

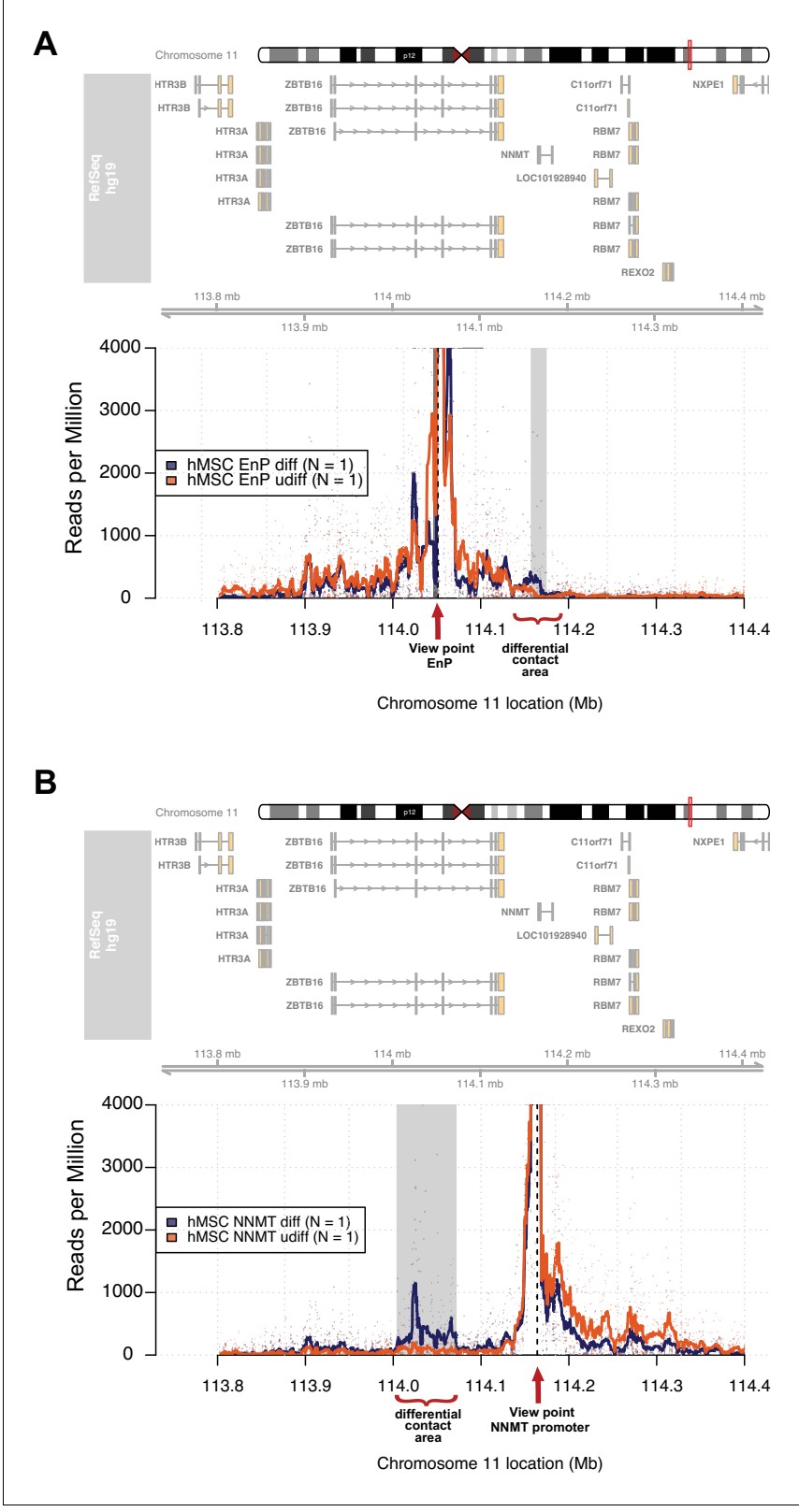

**Figure 5.** The enhancer EnP loops to the promoter of the nearest gene *NNMT* and regulates its expression. At the top, the ideograms represent the human chromosome 11. The chromosomal location of the 4C-seq profiles is indicated by the red mark. RefSeq hg19 genes are indicated by rectangles (exons) and arrowheads with connecting lines (introns) that point in the direction of transcription. (**A**) The 4C-seq contact map of the EnP

*Figure 5 continued on next page*

*Figure 5 continued*

enhancer in naive hMSCs and in pre-osteoblasts (2 days of osteogenic differentiation). The plot represents the overlay between naïve hMSCs and pre-osteoblasts. Data are displayed as reads per million (RPM). The arrow indicates the view point for depicted track and asterisks indicate contact points revealed by 4C-Seq. The plots reveal that EnP physically contacts the *NNMT* promoter and vice versa (shown in B). Gray rectangles indicate areas with higher contact frequency in pre-osteoblasts over naive hMSCs ($X^2$-test; False Discovery Rate < 0.01). (**B**) The *NNMT* promoter as a view point in 4C-Seq revealed a high frequency of contact at a 30 kb region in intron 2 and 3 of the *ZBTB16* locus after 2 days of osteogenic differentiation. The data presented here is representative of two biological replicates. Gray rectangles indicate the EnP-enhancer area with higher contact frequency in pre-osteoblasts over naive hMSCs ($X^2$-test; False Discovery Rate < 0.01).

The online version of this article includes the following figure supplement(s) for figure 5:

**Figure supplement 1.** Long range interaction map using control regions as view point revealed by 4C-Seq in hMSCs before (naive) and after induction of osteogenic differentiation (day 2).

---

obviously needs to be tightly regulated. However, our data suggest that fine-tuning of global methylation during osteogenic differentiation could partly result from small changes in SAM levels via *NNMT* expression regulated through PLZF and the intragenic enhancer EnP.

Overall, these results provide evidence for a dynamic activation and function of a lineage specific developmental enhancer EnP present within the *ZBTB16* locus, which regulates the expression of *NNMT* and thereby SAM homeostasis upon osteogenic differentiation of hMSCs.

To further characterize the effect of NNMT on osteogenic differentiation of hMSCs, we used siRNA to knock-down *NNMT* expression. We tested three different siRNA oligos to target NNMT and found all three oligos showed significant knockdown efficiency (*Figure 6—figure supplement 1*). As shown in *Figure 6* reduction in NNMT levels (panel E) inhibited the increase in *ZBTB16* and *ALPL* expression (panels F and G, respectively) normally observed during osteogenic differentiation suggesting a block in differentiation.

## PLZF is required for the recruitment of Mediator and RNA PolII at the EnP enhancer

Next, we wanted to explore the function of PLZF at the EnP enhancer and its involvement in *NNMT* expression. Therefore, we first investigated if PLZF knockdown had any influence on *NNMT* expression. Interestingly, the knockdown of PLZF negatively influenced *NNMT* gene induction (*Figure 7A, B*). This was also reflected in SAM levels, since the reduction in SAM levels previously observed during osteogenic differentiation (*Figure 6D*) was abrogated in the absence of PLZF (*Figure 7C*). We next tested if the looping between EnP and the *NNMT* promoter was affected in the absence of PLZF. We performed 4C-seq experiments in the PLZF knockdown hMSCs using the EnP enhancer element as the viewpoint in naïve and in pre-osteoblasts. To our surprise, the EnP element looped to the *NNMT* promoter with the same frequency irrespective of PLZF knockdown (*Figure 7—figure supplement 1A and B*). This suggested that PLZF does not affect the chromatin looping between EnP and the *NNMT* promoter.

Earlier, we have shown an enrichment of Mediator at EnP upon osteogenic differentiation (*Figure 4C*). Therefore, we wondered if PLZF played a role in facilitating the binding of Mediator and RNAPolII at EnP. To address this question, we next performed ChIP for MED12 and RNAPolII in the presence or absence of PLZF and induction of osteogenic differentiation. Interestingly, the recruitment of MED12 and RNAPolII at the EnP enhancer was dependent on PLZF expression upon induction of osteogenic differentiation (*Figure 7D*).

Taken together, these results suggest that PLZF influences the expression of the *NNMT* gene via recruitment of Mediator and RNAPolII at the EnP enhancer, without regulating enhancer-promoter looping per se.

## Discussion

In this study, we have shown that the induction of the *ZBTB16* gene locus encoding the transcription factor PLZF is an important event during osteogenic differentiation of hMSCs. This is in agreement with the genetic knockout of the *Zbtb16* in mice, showing severe defects in limb and axial skeleton

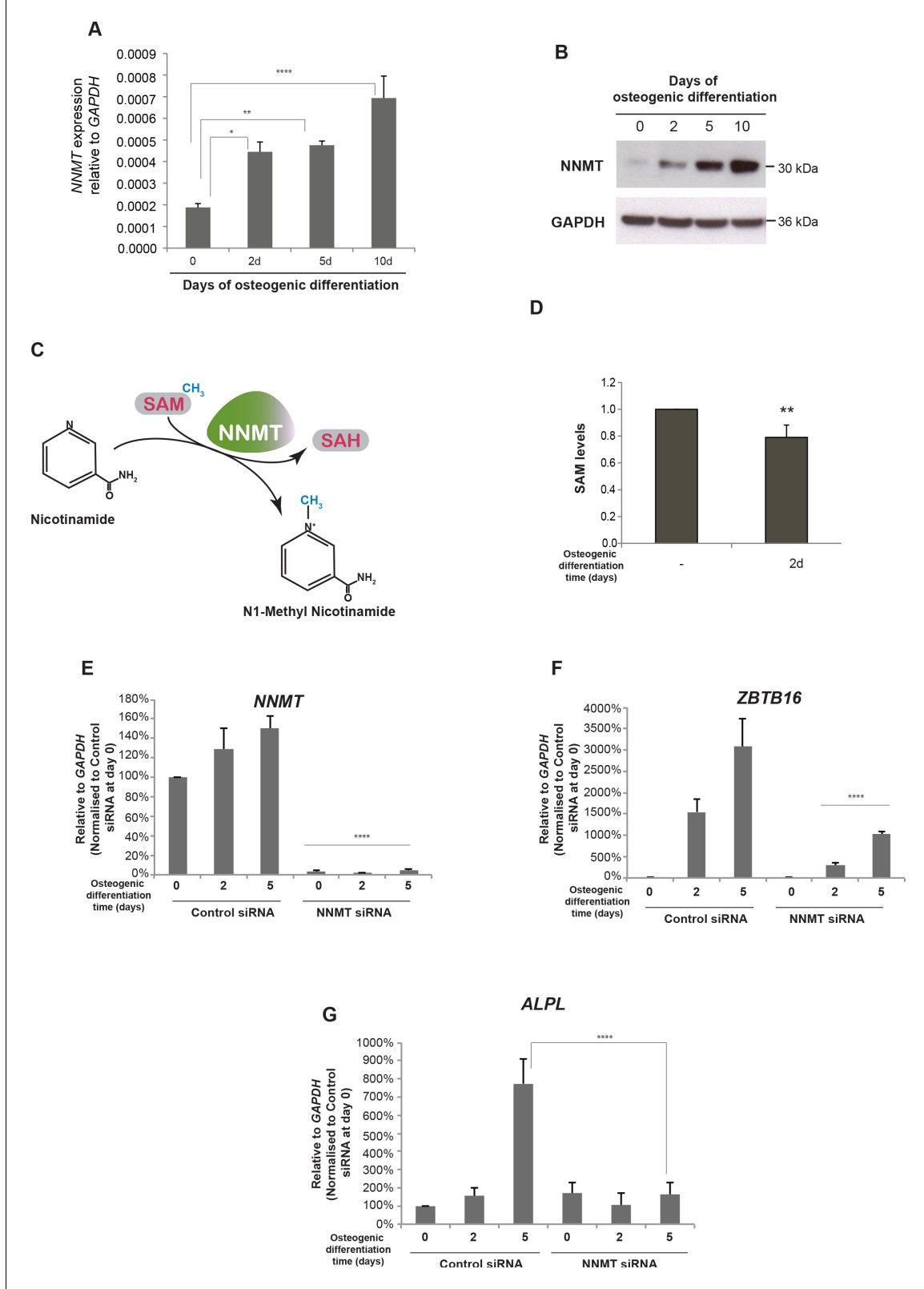

**Figure 6.** *NNMT* gene expression is induced during osteogenic differentiation and affects *ZBTB16* and *ALPL* expression. (**A–B**) *NNMT* expression during osteogenic differentiation as analyzed by RT-QPCR and western blot. RT-QPCR data represents mean ± SD from three biological replicates. *p = 0.0223, **p = 0.0011, ****p < 0.0001, significance was calculated by two-way ANOVA using Bonferroni's multiple comparisons test. (**C**) Schematic presentation of the biochemical reaction whereby NNMT transfers a methyl group from SAM (S-adenosyl methionine) to nicotinamide, producing

*Figure 6 continued on next page*

*Figure 6 continued*

1MNA (one methyl nicotinamide) and SAH (S-adenosyl homocysteine). (D) SAM levels measured by a fluorescence based assay (Mediomics) in hMSCs before and after osteogenic differentiation for 2 days. The data shown are averages of three biological replicates (p value = 0.01, two tailed t-test). (E) *NNMT* knockdown using siRNA in naive and osteogenic differentiated hMSCs, assessed by RT-QPCR. Two different siRNAs were used for knockdown of *NNMT* expression. Experiments shown are mean ± SD from three biological replicates. (F) Expression of *ZBTB16* after *NNMT* knockdown analyzed by RT-QPCR. Experiments shown are mean ± SD from two biological replicates. (G) Expression of *ALPL* after *NNMT* knockdown analyzed by RT-QPCR. Experiments shown are mean ± SD from two biological replicates. ****p < 0.0001, statistical significance was calculated by two-way ANOVA using Tukey's multiple comparisons test.

The online version of this article includes the following source data and figure supplement(s) for figure 6:

**Source data 1.** NNMT expression by RT-QPCR.
**Source data 2.** ALPL expression by RT-QPCR.
**Source data 3.** PLZF expression by RT-QPCR.
**Figure supplement 1.** (A) NNMT knockdown using three different siRNA oligos (20, 21, 22) in naïve and osteogenic differentiated (2 and 5 days) hMSCs, assessed by Q-RTPCR.

patterning (*Barna et al., 2000*). Unexpectedly, however, we found that PLZF, rather than functioning as a gene repressor by binding at promoters as described in other cellular models (*Liu et al., 2016*), mainly localized to gene enhancers in immature osteoblasts. Increased H3K27ac at PLZF-binding sites often correlated with higher expression of nearby transcripts in a PLZF-dependent manner, many of which represent protein encoding genes known to be important in osteogenic differentiation, and thereby supporting a more general role of PLZF at osteogenic lineage-specific enhancers (*Supplementary file 4*).

We also revealed the presence of an intragenic enhancer (EnP) within this locus that is repressed by PcG and H3K27me3 in naive hMSCs but becomes highly active at an early stage of osteogenic differentiation of hMSCs. Our data also demonstrate the differentiation-specific looping of EnP to its target promoter of the *NNMT* gene. This is now one of the few known examples where dynamics of enhancer promoter interaction has been described during mesenchymal stem cell differentiation (*Dixon et al., 2015*).

## PLZF marks the differentiation onset of naive hMSCs

In line with previous studies (*Djouad et al., 2014*), we observed that PLZF expression was not specific for the osteogenic lineage since induction of chondrogenic- and adipogenic differentiation of hMSCs also led to induction of *ZBTB16* expression (data not shown). These observations suggest that PLZF functions at an early state of hMSCs differentiation and promotes the transition from the naive stem cell stage to a more committed state for all three lineages. Therefore, to obtain specific gene expression patterns that characterize the three different lineages and their functions, other cell-type-specific TFs must operate together with PLZF at enhancers in a context dependent manner. Recently, the genome-wide binding profile for Runx2 was characterized in mouse pre-osteoblastic cell lines and suggested a possible link of this TF to enhancer function (*Meyer et al., 2014*; *Wu et al., 2014*). Future studies using human MSCs should be designed to reveal if RUNX2 and FOXO1 bind enhancers and cooperate with PLZF or work independently in gene regulation during osteogenic commitment. In our de novo motif analyses of PLZF-binding sites, we identified the TACAGT motif, that is identical to previously published data (*Ivins et al., 2003*). The motif is, furthermore, a recognition site for the transcription factor OSR2, implicated in proliferation of osteoblasts and in osteogenic differentiation of dental follicle cells (*Kawai et al., 2007*; *Park et al., 2015*). Whether OSR2 and PLZF compete or cooperate for binding to enhancers and if OSR2 functions together with PLZF to regulate osteogenic gene expression is not known and would require further studies. We observed that MED12 binding at the osteogenic enhancer EnP was dependent on PLZF, although no direct interaction has been observed between PLZF and MED12 (data not shown), suggesting involvement of other cofactor(s). It should be considered that, the influence of PLZF on H3K27ac at distal regulatory elements might facilitate MED12 binding at these regions which is reverted in the absence of PLZF.

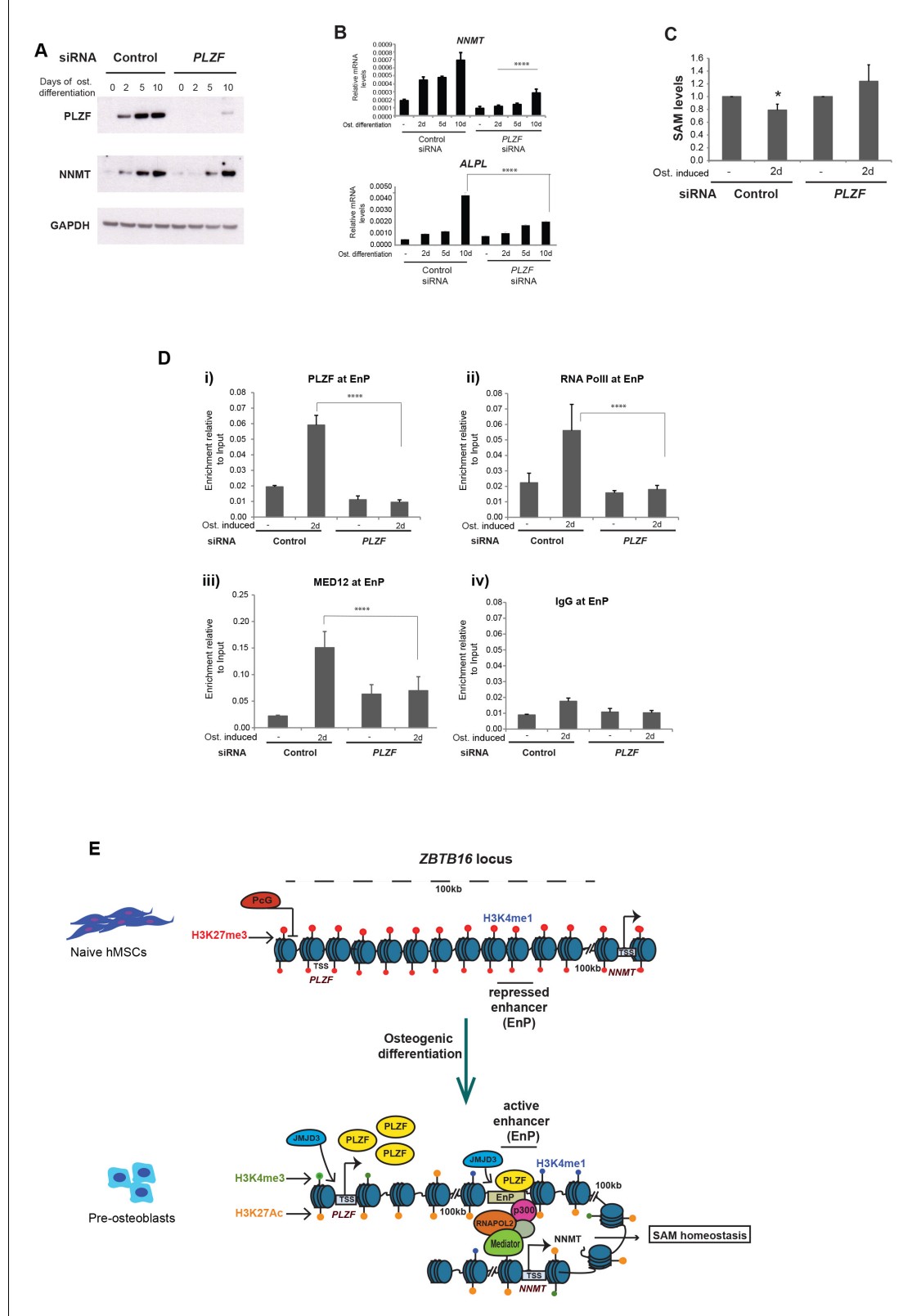

**Figure 7.** PLZF is required for MED12 and RNA PolII recruitment at the EnP enhancer.  A) Knockdown of PLZF by siRNA prevents efficient induction of the *NNMT* gene. hMSCs were transfected with siPLZF or control siRNA, left untreated or added osteogenic differentiation medium in a time course experiment. (A) Western blot showing the knockdown efficiency of PLZF and the influence on *NNMT* expression during osteogenic differentiation (western blot in A, representative blot from two biological replicates and RT-QPCR in B, the data shown are mean ± SD from three biological replicates,

*Figure 7 continued on next page*

*Figure 7 continued*

****p < 0.0001). The impact of PLZF knockdown on *ALPL* expression is shown in the lower part of the panel B. The data shown are mean ± SD from three biological replicates, ****p < 0.0001, significance was calculated by two-way ANOVA using Tukey's multiple comparisons test. (**C**) Fluorescence based SAM assay revealed that PLZF knockdown prevented the decline in SAM levels as observed in control siRNA cells (tested as significant p = 0.01 two tailed t-test) during early osteogenic differentiation (day2). The data shown here is an average of three biological replicates. (**D**) ChIP followed by QPCR for PLZF, RNA PolII, and MED12 at the EnP element before and after PLZF knockdown in hMSCs. Osteogenic differentiation was induced one day after siRNA transfection and cells were fixed for ChIP two days later. The data represents two biological replicates and show averages of triplicate values from QPCR ± SD. ****p < 0.0001, significance was calculated by two-way ANOVA using Tukey's multiple comparisons test. (**E**) Model to show that the *ZBTB16* locus is repressed by Polycomb protein complexes (PcG) and marked by H3K27me3 in naive hMSCs. Upon induction of osteogenic differentiation, the *ZBTB16* locus gets derepressed by losing PcG binding, and H3K27me3 through JMJD3 recruitment, gain H3K27ac which eventually results in high expression of PLZF. The intragenic enhancer 'EnP' gets exposed, gains H3K27ac and H3K4me1 histone marks and binds PLZF as well as P300 and the Mediator complex. Subsequently, the EnP element loops to the promoter of the *NNMT* gene (100 kb downstream) and induce its expression and as a consequence regulates SAM homeostasis during osteogenic differentiation of hMSCs.

The online version of this article includes the following source data and figure supplement(s) for figure 7:

**Source data 1.** ChIP for enhancer binding proteins in hMSCs, in the absence of PLZF.
**Figure supplement 1.** 4C-sequencing map revealing that the PLZF knockdown (using siRNA) did not affect the contact frequency between the EnP element and *NNMT* promoter observed during early osteogenic differentiation (day 2).

## A latent developmental enhancer EnP within the *ZBTB16/PLZF* locus

EnP that lies within the *ZBTB16* locus represents a latent PcG-repressed developmental enhancer in naive hMSCs. Upon induction of osteogenic differentiation, H3K27me3 was removed from EnP possibly through JMJD3 recruitment and there was a subsequent gain of chromatin marks characterizing active enhancers (H3K4me1 and H3K27ac) and gene regulatory factors such as Mediator, p300 and PLZF. Intriguingly, a reporter assay in which EnP was integrated in the genome of hMSCs cells showed that EnP functions only upon induction of differentiation. This suggests that the EnP enhancer element in the context of a GFP reporter also require lineage-specific factor(s) such as PLZF for its activity.

The region of the *ZBTB16* locus that we found to be in contact with the promoter of the *NNMT* gene covered an area around 50 kb and included introns 3–5 which showed a strong enrichment for H3K27ac and H3K4me1 upon osteogenic differentiation and gained PLZF binding at several positions. These characteristics suggest that this *ZBTB16* intragenic region could represent a cluster of several active enhancers (*Whyte et al., 2013*) that likely binds other TFs besides PLZF. Since increased expression of PLZF was also observed during induction of chondrogenic and adipogenic differentiation (data not shown), we assume that EnP might be functional in these lineages as well. However, whether all potential intragenic enhancers in the *ZBTB16* locus would be regulated similarly between the three different cell lineages and contact similar or different target promoters would need further investigation.

## Regulation of SAM levels through *NNMT* expression and activity

We found that EnP loops to the *NNMT* promoter and correlated with induced expression of the *NNMT* gene during osteogenic differentiation, and that NNMT expression is required for the differentiation of hMSCs. The enzyme NNMT regulates nicotinamide (vitamin B3) levels through methylation, using SAM as a methyl donor (*Figure 6C*). NNMT is overexpressed in many cancers (*Sartini et al., 2013*; *Ulanovskaya et al., 2013*) and can result in an altered epigenetic state. This has been proposed to happen due to draining of methyl groups from the methionine cycle leaving very stable 1MNA and SAH byproducts, thereby changing the methylome of cancer cells (*Shlomi and Rabinowitz, 2013*; *Ulanovskaya et al., 2013*). Based on our data, it is tempting to speculate that the *ZBTB16* intragenic enhancer EnP and PLZF, helps to fine tune global methylation patterns through transcriptional regulation of the *NNMT* gene during osteogenic commitment of progenitor cells. These important aspects require further detailed investigations.

In conclusion, our results identify the transcription factor PLZF as a novel component that marks the enhancer landscape in hMSCs during osteogenic differentiation, acting as a positive regulator of enhancer function. Importantly, we find that the derepression of the *ZBTB16* locus, besides causing increased PLZF expression, furthermore, exposes an intragenic developmental enhancer EnP that

contact the promoter of the *NNMT* gene through chromatin looping and affects SAM homeostasis in osteogenesis (see model *Figure 7E*).

# Materials and methods

## Key resources table

| Reagent type (species) or resource | Designation | Source or reference | Identifiers | Additional information [RRID] |
|---|---|---|---|---|
| Gene (*Homo sapiens*) | ZBTB16 (zinc finger and BTB domain containing 16provided by HGNC) | HGNC:HGNC: 12930 | Gene ID: 7704, updated on 18-Nov-2018 | |
| Gene (*H. sapiens*) | NNMT (nicotinamide N-methyltransferase provided by HGNC) | HGNC:HGNC: 7861 | Gene ID: 4837, updated on 11-Nov-2018 | |
| Cell line (*H. sapiens*) | Human Mesenchymal Stem Cells (HMSC) | Bone Marrow derived, from Lonza | Lonza, Cat. No. PT-2501 | |
| Recombinant DNA reagent | PCR8 TOPO TA vector | Invitrogen | | |
| Recombinant DNA reagent | pSINMIN lentiviral 'enhancer reporter vector' | Johanna Wysoca's lab | (*Rada-Iglesias et al., 2011* | |
| Software, algorithm | Easeq | https://easeq.net | Nature Structural and Molecular Biology Volume 23 No 4 (April 2016), 349–357 | |
| Chemical compound, | Lipofectamin 2000 | Invitrogen | | |
| Chemical compound, | StemPro osteogenesis | Lonza | Gibco A10072-01 | |
| Sequence-based reagents | Mission siRNA oligos ZBTB16 SiRNA | Sigma | SASI_Hs01_00148557 | |
| Sequence-based reagents | Mission siRNA oligos ZBTB16 SiRNA | Sigma | SASI_Hs01_00148556 | |
| Sequence-based reagents | Mission siRNA oligos ZBTB16 SiRNA | Sigma | SASI_Hs01_00148555 | |
| Sequence-based reagents | Mission siRNA oligos NNMT SiRNA | Sigma | SASI_Hs01_00209920 | |
| Sequence-based reagents | Mission siRNA oligos NNMT SiRNA | Sigma | SASI_Hs01_00209921 | |
| Sequence-based reagents | Mission siRNA oligos NNMT SiRNA | Sigma | SASI_Hs01_00209922 | |
| Antibody | PLZF (Rabbit polyclonal) | Santa Cruz | sc-22839 RRID: AB_2304760 | WB-1:600 ChIP-5μg |

*Continued on next page*

*Continued*

| Reagent type (species) or resource | Designation | Source or reference | Identifiers | Additional information [RRID] |
|---|---|---|---|---|
| Antibody | NNMT (Rabbit polyclonal) | Abcam | ab58743 RRID: AB_881715 | WB (1:1000) |
| Antibody | GAPDH (Mouse monoclonal) | Abcam | ab8245 RRID: AB_2107448 | WB (1:10,000) |
| Antibody | CyclinE (Mouse monoclonal) | Abcam | ab3927 RRID: AB_304167 | WB (1:1000) |
| Antibody | pRB2 (Rabbit polyclonal) | Santa Cruz | sc317 RRID: AB_632093 | WB (1:1000) |
| Antibody | anti-H3K27me3 (Rabbit monoclonal) | Cell Signalling | 9733 RRID: AB_2616029 | ChIP-5µl |
| Antibody | anti-SUZ12 (Rabbit monoclonal) | Cell Signalling | 3737 RRID: AB_2196850 | ChIP-5µl |
| Antibody | Anti-RNF2 (RING1B) (Rabbit polyclonal) | Home made | Peptide 'NAST' RRID: AB_2755047 | ChIP-5µg |
| Antibody | anti-H3K4me3 (Rabbit monoclonal) | Cell Signalling | 9751 RRID: AB_2616028 | ChIP-5µl |
| Antibody | anti-H3K27ac (Rabbit polyclonal) | Abcam | ab4729 RRID: AB_2118291 | ChIP-5µg |
| Antibody | anti-H3K4me1 (Rabbit polyclonal) | Abcam | ab8895 RRID: AB_306847 | ChIP-5µg |
| Antibody | anti-H3K4me1 (Rabbit monoclonal) | Cell Signalling | 5326S RRID: AB_10695148 | ChIP-5µl |
| Antibody | anti-H3K36me3 (Rabbit monoclonal) | Cell Signalling | 4909 RRID: AB_1950412 | ChIP-5µl |
| Antibody | P300 (Rabbit polyclonal) | Santa Cruz | sc-585 RRID: AB_2231120 | ChIP-5µg |
| Antibody | RNA POLII (Rabbit polyclonal) | Santa Cruz | sc 9001X RRID: AB_2268548 | ChIP-5µg |
| Antibody | MED1/TRAP220 (Rabbit polyclonal) | BETHYL | A300-793A RRID: AB_577241 | ChIP-5vg |
| Antibody | MED12 (Rabbit polyclonal) | BETHYL | A300-774A RRID:AB_669756 | ChIP-5µg |
| Antibody | JMJD3 (KDM6B) (Rabbit polyclonal) | home made | Peptide 'KAKA' RRID: AB_2755046 | ChIP-5µg |

*Continued on next page*

*Continued*

| Reagent type (species) or resource | Designation | Source or reference | Identifiers | Additional information [RRID] |
|---|---|---|---|---|
| Commercial assay or kit | Bridge-It S-Adenosyl Methionine (SAM) Fluorescence Assay Kit | Mediomics | 1-1-1003A (50 measurements) | |
| Commercial assay or kit | DNeasy Blood and tissue kit | QIAGEN | 69504 | |
| Commercial assay or kit | RNeasy Plus Mini kit | QIAGEN | 74106 | |
| Commercial assay or kit | TaqMan Reverse Transcription Reagents | Applied Biosystems | N808-0234 | |
| Commercial assay or kit | Fast SYBR Green Master Mix | Applied Biosystems | 4385612 | |
| Commercial assay or kit | affymetrix gene chips | Affymetrix | HT_HG-U133_Plus_PM | |
| Commercial assay or kit | ChIP seq DNA sample preparation kit | Illumina | Catalog IDs: IP-102–1001 | |

## Human mesenchymal stem cells (hMSC) culture and differentiation

Bone-marrow-derived primary human mesenchymal stem cells (hMSC) were purchased from Lonza (Lonza, Cat. No. PT-2501). Cell purity and their ability to differentiate into osteogenic, chondrogenic and adipogenic lineages were tested by Lonza. Cells were positive for CD105, CD166, CD29, and CD44. Cells tested negative for CD14, CD34 and CD45. Cells were cultured according to manufacturer's instructions using mesenchymal stem cell growth medium (MSCGM) (Lonza MSCGM: PT-3238) supplemented with one MSCGM SingleQuots (PT-4105). Differentiation was induced using StemPro osteogenesis (Gibco A10072-01), chondrogenesis (Gibco A10071-01) or adipogenesis (Gibco A10070-01) differentiation kit as per manufacturer's instructions. Cells were harvested at the indicated time points for ChIP, RNA, 4C-seq and protein using appropriate buffers as described below.

## PLZF knockdown using siRNA

Mission siRNA oligos were ordered from Sigma. Three different oligos were tested and sequences are provided in the *Supplementary file 9*.

The hMSCs were reverse transfected with siRNA oligos using Lipofectamin 2000 and Optimem media (Gibco). Briefly, 600 pmol of siRNA (in 500 µl Optimem was mixed with 10 µl Lipofectamin 2000 (Invitrogen) in 500 µl Optimem. After 15 min of incubation at room temperature, $1 \times 10^6$ cells were mixed in the transfection mix and incubated for another 10 min. Cells in the transfection mix were seeded in dishes (15 cm, Nunc cell culture) with normal HMSCB media. Osteogenic differentiation was induced 24 hr after transfection and cells processed for analyses at indicated time points.

## Chromatin immunoprecipitation (ChIP) assay

Cells were processed for ChIP as previously described with slight modifications (*Dietrich et al., 2012*). Briefly, cells were fixed for 10 min in culture media containing 1% formaldehyde and were processed for ChIP as previously described with slight modifications (*Dietrich et al., 2012*). Briefly, formaldehyde fixed chromatin was sonicated in IP buffer (IP buffer = 2 volumes of SDS Buffer: 1 vol Triton Dilution Buffer; SDS Buffer- 50 mM Tris-HCl, (pH8.1), 100 mM NaCl, 5 mM EDTA, (pH 8.0), 0.2% (w/v) NaN$_3$, 2% (w/v) SDS; Triton Dilution Buffer-: 100 mM Tris-HCl, (pH 8.6), 100 mM NaCl, 5 mM EDTA, (pH 8.0), 0.2% (w/v) NaN$_3$ 5.0% (v/v) Triton X-100) using a Branson Sonifier (4 cycles of 30

s each at 22% of max amplitude). Twenty µg DNA (sonicated chromatin) was used for each ChIP in IP buffer. Antibodies used: Rabbit general IgG (DAKO), rabbit monoclonal anti-H3K27me3 (Cell Signalling # 9733, Rabbit monoclonal,), anti-SUZ12 (Cell Signalling # 3737 Rabbit monoclonal), RING1B (produced in our lab), H3K4me3 (Cell Signalling # 9751, Rabbit monoclonal), H3K27ac (Abcam # ab4729), H3K4me1 (Abcam # 8895), (Cell Signalling # 5326S, Rabbit monoclonal used for ChIP-seq), H3K36me3 (Cell Signalling # 4909, Rabbit monoclonal), PLZF (Santa Cruz # sc-22839), P300 (Santa Cruz # sc-585), RNA POLII (Santa Cruz # sc 9001X), MED1/TRAP220 (BETHYL # A300-793A), MED12 (BETHYL # A300-774A), JMJD3 (produced in our lab)). Samples were first incubated with antibodies overnight at 4°C by end-over-end rotation and subsequently immune-complexes were enriched using incubation with protein A-Sepharose beads for 3 hr. After washing the samples were de-crosslinked overnight at 68°C (shaking) in ChIP Elution Buffer (20 mM Tris-HCl (pH7.5), 5 mM EDTA (pH 8.0), 50 mM NaCl, 1% (w/v) SDS, 50 µg/ml Proteinase K). Finally, enriched DNA was purified using a Qiagen PCR purification kit. The ChIPs were validated at Polycomb target genes by QPCR.

## ChIP sequencing

ChIP DNA from three parallel ChIPs were pooled or individual ChIPs from three biological replicates were used and 10 ng each was used for making ChIP-seq libraries. The libraries were prepared using 'ChIP seq DNA sample preparation kit' from Illumina following manufacturer's instructions. Individual samples were runsequenced in a single lanes on HiSeq2000 (Illumina). H3K27ac samples were or multiplexed (using NEB kit) and run on a Illumina Genome Analyzer IIx, HiSeq2000 (Illumina), or NextSeq500 as single end sequencing.

## 4C-Sequencing

Templates for 4C were prepared as described previously (*van de Werken et al., 2012a*; *van de Werken et al., 2012b*). The enzymes used were four base pair cutters DpnII and Csp6I.

Osteogenic induced (2 days; preosteoblasts) or naive hMSCs ($10 \times 10^7$ cells per group) were harvested and fixed in 2% (v/v) formaldehyde in PBS, 10% (v/v) FBS, for 10 min with rotation at room temperature. The procedure was continued as described earlier (*van de Werken et al., 2012a*).

The primer sequences are given in the Table with all primers. The final PCR amplification was performed using 3 µg of 4C DNA, and PCR conditions were 98°C (2 min), 98°C (15 s), 58°C (1 min), 72°C (3 min), repeated for 30 cycles and final amplification for 7 min at 72°C using Phusion High Fidelity DNA Polymerase (Thermo Scientific).

## S-adenosyl methionine (*SAM*) assay

SAM was measured using the Bridge-It S-Adenosyl Methionine (SAM) Fluorescence Assay Kit from Mediomics according to the manufacturer's instructions (Mediomics, LLC, St. Louis, Missouri, U.S. A.). Osteogenic differentiation of hMSCs was induced and cells were harvested after 2 days together with naïve cells and processed for SAM assay. The procedures was followed as described in the assay kit manual.

## RT-PCR, microarray and RNA sequencing

Total RNA representing the indicated time points were purified from hMSCs using RNeasy Plus Mini kit (QIAGEN 74106), and cDNA was generated by RT-PCR using the TaqMan Reverse Transcription Reagents (Applied Biosystems#N808-0234). Quantifications were done using the Fast SYBR Green Master Mix (Applied Biosystems#4385612) and an ABI Step One Plus instrument. *GAPDH* was used for normalization. The sequences of the primers used are listed in the *Supplementary file 9* with primers. The RNA samples at days 0, 2, 5, and 10 of osteogenic differentiation (in three biological replicates) were sent for microarray hybridization using affymetrix gene chips (HT_HG-U133_Plus_PM).

For RNA sequencing, samples were processed using a True Seq RNA Sample preparation kit (Illumina # 15025062) following the manufacturer's instructions. Samples were multiplexed and sequenced in HiSeq2000.

## Enhancer activity reporter assay

The pSINMIN lentiviral 'enhancer reporter vector' was kindly provided by Johanna Wysoca's lab (*Rada-Iglesias et al., 2011*). The enhancer fragment (EnP) (corresponding to the biggest PLZF peak

within the *ZBTB16* locus) or control element region without a PLZF peak within the *ZBTB16* locus (CtE) were PCR amplified from genomic DNA (gDNA) isolated from hMSCs using primers described in the primer Table and cloned into the PCR8 TOPO TA vector (Invitrogen). The relevant fragments were cut out from the TOPO vector using EcoRI sites and subsequently cloned into the pSINMIN vector using EcoRI sites. Positive clones were confirmed by sequencing.

Lenti-virus were produced by transfecting 20 μg reporter plasmid together with 15 μg PAX8 and 6 μg of VSV in 293FT cells seeded at a density of $5 \times 10^6$ cells per 15 cm dish (Nunc cell culture) using the $CaPO_4$ method. 48 hr following transfection and media change the supernatant containing viral particles were collected and filtered through 0.45 μm filters. Polybrene was added to the supernatant 8 μg/ml and hMSCs were infected with virus overnight. Cells were trypsinised and reseeded 48 hr later and Puromycin (0.5 μg/ml) was added for selection. Two days later, osteogenic differentiation was induced and cells were harvested after 2 days for flowcytometric analyses or gDNA isolation. Genomic DNA was isolated using the DNeasy Blood and tissue kit (QIAGEN: 69504). Quantitative PCR was performed using GFP primers described in the primer Table. QPCR on the *HOXD* locus was used for normalization. Untransduced hMSCs were used as a control.

## Western blotting

Western blotting was performed according to standard procedures. The cells were lysed in high salt lysis buffer (50 mM Tris-HCl (pH 7.2), 300 mM NaCl, 0.5% (w/v) Igepal CA-630, 1 μg/ml aprotinin, 1 μg/ml leupeptin, 1 mM PMSF, 1 mM EDTA (pH 8.0), 50 mM NaF, 20 mM β-glycerophosphate) followed by a brief sonication on a Branson Sonifier with a 2 mm probe (10% of max amplitude for 2 s on ice). Samples were left on ice for 30 min, then centrifuged at 22,000 g for 15 min. Ten percent Tris-glycine or 4–12% Bis-Tris SDS-PAGE gels (Invitrogen) were used to separate proteins for analyses. Seablue Plus two prestained marker (Invitrogen) was used as a molecular weight standard. Blotting was performed using the following antibodies: PLZF (Santa Cruz # sc-22839), NNMT (Abcam # ab58743), GAPDH (Abcam # ab8245), cyclinE (mab, HE12), pRB2 (Santa Cruz, sc317).

## Alizarin red staining

Naive or osteogenic induced hMSCs were washed two times with PBS followed by fixation in 96% ethanol for 15 min at room temperature. Alizarin red solution 1% (w/v, dissolved in milliQ water) was added to the fixed cells and incubated for 60 min at room temperature with gentle rotation. Finally, cells were carefully (not to lose precipitates) washed three times with water and dried. Pictures were taken with a Leica camera (DFC295) fitted to a microscope at 20X magnification.

## Bioinformatic analyses

### ChIP sequencing

Reads were trimmed for low-quality nucleotides and mapped to the hg19 human genome sequence using Bowtie v0.12.7 (*Langmead et al., 2009*) using the parameters '-S -m 1'. When nothing else is mentioned, subsequent processing, signal quantitation, peak-finding, distance calculation, normalization, clustering and visualization was done using EaSeq and default settings (http://easeq.net *Lerdrup et al., 2016*). Values in 2D-histograms in tracks or heatmaps were counted and normalized to fragments per million reads per kbp. The number of fragments was derived from the count by dividing it with (1 + DNA fragment size / bin size). Aligned datasets were filtered for multiple reads at the same position likely to occur from PCR amplification. The reads were extended to a DNA fragment size of 300 bp. Genes subject to differential H3K27me3 levels were identified as follows: Read counts from two biological H3K27me3 ChIP-seq replicates from day 0 (naïve hMSCs) and day 10 (immature osteoblast) samples were quantified within 5five kbp windows surrounding 30,716 nonredundant TSS in Refseq Hg19 gene annotation data (*O'Leary et al., 2016*), which were downloaded from the refflat table at UCSC (*Kent et al., 2002*). TSSes were filtered for low abundance by requiring a read count >=15 from each sample and >=100 for all four samples collectively. The 8,222 TSSes meeting these criteria were analysed for fold change and significance using DESeq2 (*Love et al., 2014* ). For 2D-histograms of ChIP-seq data, Tthe number of reads overlapping with an area of +/-1 kbp of transcript start to end for H3K27me3 and ±1 kbp of TSS for H3K27ac and H3K4me3 were quantified. Each data set was quantile normalized in order to compensate for variations in ChIP efficiency between the samples. Regions with low H3K27me3 levels (<0.5) or low

H3K4me3 (<5) in both samples were filtered and two fold changes in each histone mark was calculated (day10/day0 (naïve)). Regions with gain or loss in each histone mark were calculated by selecting the population with twofold change or higher in either direction. Values in 2D-histograms in tracks or heatmaps were counted and normalized to fragments per million reads per kbp. The number of fragments was derived from the count by dividing it with (1 + DNA fragment size/bin size).

## Peak-finding

Specific PLZF peaks were identified using the IgG library data set as negative control. The procedure resembles that of MACS with some modifications (*Zhang et al., 2008*). Each dataset was divided into 100 bp windows, and the reads within each window scanned genome-wide. A normalization coefficient (NCIS) serving to normalize the background levels of the two datasets was analyzed in accordance with Liang and Keles, (*Liang and Keleş, 2012*). Window size was manually set to 100 bp. Global thresholds were calculated based on a Poisson Distribution using the genome-wide average number of reads in the windows and a p-value of 1E-05. Adaptive thresholds were modeled the following way: The average number of negative control reads in areas corresponding to 10x, 50x and 250x window size (100 bp) was calculated for each position. This number was used as lambdas for Poisson distributions and thresholds that matched a p-value of 1E-05 were calculated. The most conservative threshold was chosen from the three local thresholds of the control and the global threshold of the sample. Thresholds from the negative control were scaled according with the NCIS normalization factor. The position and statistics of windows passing the most conservative threshold, and having a NCIS-normalized log2-fold Sample/Control-ratio above 2, as well as less than a 3:1 difference between the signal on the plus and minus strands were extracted into a separate list, and the entire procedure was done four times where the windows were shifted 25 bp each time. Windows within 100 bp of each other and overlapping windows were merged. For each region in the resulting the borders were refined by sliding a window of 100 bp from one window-size upstream to downstream of the temporary border. The exact position where the number of samples reads within the window fell below the threshold was defined as the new border of that region. Shoulders were excluded at values below μ +2 SD. After border refinement and peak-merging, peaks were positively selected for an FDR value of 1E-05 Benjamini or better and minimum a NCIS normalized log2 fold difference of 2. To focus on those PLZF peaks from day 10 that were unique for differentiated cells, those peaks that had more than 0.76 fragments/kbp/M in the day 0 (naïve hMSCs) sample (quantified ±500 bp of the peak center) were excluded.

## Motif analysis

De novo motif analysis was done using the MEME suite at http://meme-suite.org (*Bailey et al., 2009*) and the MEME-ChIP tool set (*Machanick and Bailey, 2011*) to search for new motifs in discriminative mode. The central enrichment of output from MEME and DREME (*Bailey, 2011*) was Central enrichment was visualized using the integrated Centrimo tool (*Bailey and Machanick, 2012*). Negative control loci were generated using EaSeq (*Lerdrup et al., 2016*), and the tool 'Controls' to make a set of control loci that on population level matched the PLZF peaks in distance and orientation to the closest TSS. Fasta files of sequences from the two sets of regions were generated using the 'Get sequences' tool in EaSeq.

## RNA sequencing

Data received from RNA sequencing was analyzed by using Genomatix software (Expression Analysis for RNASeq Data, GGA) (Genomatix Software GmbH, Germany) as per their guidelines. Normalized gene expression was calculated using following formula (by the software):

NE = c * #reads / (#reads * length) where NE is the normalized expression or enrichment value,

#reads: the reads (sum of base pairs) of falling into either the transcript or the cluster region,

#reads: all mapped reads (in base pairs),length: the transcript or cluster length in base pairsand c a normalization constant set to $10^7$.

## ChIP-seq, RNA-seq, and microarray integration

Significantly regulated genes from the microarray analysis were imported as a Regionset in EaSeq ((*Lerdrup et al., 2016*) and translated into genomic coordinates using the 'Find coordinates in an

already loaded geneset file' option based on gene names and a Refseq (*O'Leary et al., 2016*) hg19 annotation file downloaded and imported as Geneset on October 30 2018 from the UCSC table browser (*Karolchik et al., 2004*). Of 1,286 imported genes the gene names corresponding coordinates were not found in 187 cases, which were depicted as white in the heatmaps. The list of upregulated genes at d2 in *Supplementary file 1* was imported similarly, resulting in 441 coordinates with non-zero read counts in the control. Heatmaps were made using the 'ParMap' plot type, and the color scale based on http://www.ColorBrewer.org (*Harrower and Brewer, 2003*). Beeswarm plotting and Mann-Whitney U-testing was done using R (*R Development Core Team, 2014*) and the beeswarm package (*Eklund, 2016*). Similarly, the list of transcript coordinates and mRNA quantities derived from the RNA-seq analysis was imported into EaSeq and used for quantitation of ChIP-seq signal at TSS of the gene encoding each transcript.

### Colocalization of PLZF peaks and gene expression

To visualize expression changes of genes near PLZF peaks, For each peak the the nearest list of transcripts was identified from the imported RNA-seq analysis using the 'Colocalize'-tool was searched for the one with the closest transcript and the fold change in normalized expression of this transcript was used for visualization.

### 4C-Sequencing

The 4C-seq mapping and normalization was carried out using the mapping and normalization strategy as previously described (*van de Werken et al., 2012b*). In brief, 4C-seq reads were demultiplexed and the primers sequences without the first restriction enzyme recognition site, were removed from the reads. Subsequently, the trimmed reads were mapped to a database with in silico digested genomic fragment-ends of the human reference genome build hg19. The view point, the non-cut fragment-end and the self-ligation fragment-end were discarded. After linear interpolation of the center of the fragment-ends, the distribution of the blind-fragment reads and the non-blind fragment reads were quantile normalized using R's limma package (*Smyth, 2005*) . Furthermore, the 4C-seq contact frequencies within the locus of interest were normalized to its library size. The 4C-seq contact frequencies were used to calculate the trimmed mean (10%) with a running window size of 21 fragment-ends. Since 4C-seq contact frequencies have a PCR amplification bias, we carried-out a non-parametric approach by ranking each fragment-end and applying a $\chi^2$-test on the summed rank position for both samples on each running window. Subsequently, we determined the sample with the highest contact frequencies per window and corrected for multiple hypothesis testing using the Benjamini-Hochberg method (*Benjamini and Hochberg, 1995*). The R statistical package version 3.1.1 was used for the statistical calculations and for generating the 4C-seq plots (*R Development Core Team, 2014*). The R Gviz package was used for plotting the annotation data (*Hahne et al., 2018*).

## Acknowledgements

We thank Joanna Wysocka for providing the lentiviral GFP reporter vector (pSINMIN) and Prof. Carsten Muller-Tidow for critical reading and discussion on the manuscript. Thanks to the members of Hansen group for support and fruitful discussions, and Kathryn Wattam for excellent technical assistance. HJGvdW was supported by a Zenith grant (93511036) from the Netherlands Genomics Initiative (NGI). SAS was supported by FSS (Danish Research Council). KH was supported by the Danish National Research Foundation (DNRF82) and through a center grant from the Novo Nordisk Foundation (NNF17CC0027852). KHA was supported by the Lundbeck Foundation and the Danish National Research Foundation.

## Additional information

### Funding

| Funder | Grant reference number | Author |
| --- | --- | --- |
| Forskerakademiet | 09-073526 | Shuchi Agrawal Singh |

| Novo Nordisk | NNF17CC0027852 | Kristian Helin |
|---|---|---|
| Danish National Research Foundation | DNRF 82 | Kristian Helin<br>Klaus Hansen |
| Memorial Sloan-Kettering Cancer Center | Support Grant NIH P30 CA008748 | Kristian Helin |
| Lundbeckfonden | R34-A3479 | Klaus Hansen |

The funders had no role in study design, data collection and interpretation, or the decision to submit the work for publication.

### Author contributions
Shuchi Agrawal Singh, Conceptualization, Data curation, Formal analysis, Funding acquisition, Validation, Investigation, Visualization, Methodology, Writing—original draft, Project administration, Writing—review and editing; Mads Lerdrup, Software, Formal analysis, Writing—review and editing; Ana-Luisa R Gomes, Methodology; Harmen JG van de Werken, Jens Vilstrup Johansen, Robin Andersson, Albin Sandelin, Formal bioinformatic analysis; Kristian Helin, Visualization, writing-review, Provided critical insights throughout the study and gave comments to the manuscript; Klaus Hansen, Conceptualization, Resources, Supervision, Funding acquisition, Writing—original draft, Project administration, Writing—review and editing

### Author ORCIDs
Shuchi Agrawal Singh (iD) https://orcid.org/0000-0001-8632-4556
Mads Lerdrup (iD) http://orcid.org/0000-0002-7730-8973
Harmen JG van de Werken (iD) https://orcid.org/0000-0002-9794-1477
Albin Sandelin (iD) https://orcid.org/0000-0002-7109-7378
Kristian Helin (iD) https://orcid.org/0000-0003-1975-6097
Klaus Hansen (iD) https://orcid.org/0000-0001-9657-8816

### Decision letter and Author response
Decision letter https://doi.org/10.7554/eLife.40364.sa1
Author response https://doi.org/10.7554/eLife.40364.sa2

## Additional files

### Supplementary files
• Supplementary file 1. Microarray-Genes-differentially regulated (Osteogenic differentiation induced Vs Naive).

• Supplementary file 2. K4me3Gain-ExpressionGain-K27meLoss.

• Supplementary file 3. K27me3 changes.

• Supplementary file 4. PLZF-Peaks.

• Supplementary file 5. PLZF peaks-Expression of nearest transcript and H3K27ac changes.

• Supplementary file 6. FANTOM5-enahncers-overlapping to PLZF peak+−1 kb.

• Supplementary file 7. PLZF-dependent K27ac.

• Supplementary file 8. Differentially expressed transcripts/genes from RNA seq-(PLZF-kd 2 days diff-Vs Control Si 2 days diff, OR 2 days differentiation Vs naive).

• Supplementary file 9. Oligo sequences.

• Transparent reporting form

### Data availability
Sequencing data have been made available via GEO under accession number GSE125168. All other data generated or analysed during this study are included in the manuscript and supporting, source files: Supplementary tables associated with Figure 1, Figure 2 and Figure 3 are uploaded with the submission. CAGE-derived enhancer candidates were taken from http://slidebase.binf.ku.dk/.

The following dataset was generated:

| Author(s) | Year | Dataset title | Dataset URL | Database and Identifier |
|---|---|---|---|---|
| Shuchi Agrawal Singh | 2019 | PLZF targets developmental enhancers for activation during osteogenic differentiation of human mesenchymal stem cells | https://www.ncbi.nlm.nih.gov/geo/query/acc.cgi?acc=GSE125168 | NCBI Gene Expression Omnibus, GSE125168 |

The following previously published dataset was used:

| Author(s) | Year | Dataset title | Dataset URL | Database and Identifier |
|---|---|---|---|---|
| FANTOM Consortium and the RIKEN PMI and CLST (DGT) | 2014 | A promoter-level mammalian expression atlas | https://www.ebi.ac.uk/ena/data/view/DRA000991 | European Nucleotide Archive, DRA000991 |

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
