## [Decision Letter]

[Editors’ note: in a previous version of this study, the authors were asked to provide a plan for revisions before the editors issued a final decision. What follows is the editors’ letter requesting such plan.]

Thank you for submitting your article "PLZF targets developmental enhancers for activation during osteogenic lineage commitment of human mesenchymal stem cells" for consideration by eLife. Your article has been reviewed by three peer reviewers, and the evaluation has been overseen by a Reviewing Editor and Jessica Tyler as the Senior Editor. The reviewers have opted to remain anonymous.

The reviewers have discussed the reviews with one another and the Reviewing Editor has drafted these remarks to emphasize the work that we feel must be completed for this manuscript to be considered for publication. Given the substantial additional work the reviewers feel is essential, we ask that you respond with a detailed plan of the experimental work you would be prepared to undertake and an estimate of the time it would take to complete these tasks. The Editor and reviewers will consider your response and issue a binding recommendation. Of course, if you are unable to complete these tasks in a reasonable time, we would understand if you choose to withdraw this manuscript from further consideration at this time.

The manuscript from Singh et al. reports on genomic roles of PLZF during lineage commitment of human mesenchymal stem cells and evidence for a latent enhancer within the *ZBTB16/PLZF* locus that is developmentally induced and regulates expression of the *NNMT* gene to influence cellular levels of SAM. The studies are of general interest with respect to understanding mechanisms of osteogenic differentiation. The authors demonstrate that PLZF is strongly induced at an early time point during osteogenic differentiation, coinciding with loss of the repressive H3K27me3 mark and gain of H3K27ac. ChIP-seq studies indicate that PLZF primarily binds to intra- and intergenic regions with features of enhancers, many of which exhibit increases in H3K27ac during differentiation and are near genes that are enriched for relevant functional annotations. Chromatin conformation capture (4C) experiments are performed indicating an interaction of the latent enhancer within the *ZBTB16* locus and the *NNMT* gene, which is not dependent on PLZF induction. Induction of the expression of *NNMT*, which catalyzes the conversion of nicotinomide to N1-methyl nicotinomide in a SAM-dependent manner, results in a modest but significant reduction of SAM levels. Knockdown of the *NNMT* using an siRNA results in a striking block of induction of *ZBTB16* and ALPL mRNA during osteogenesis. The authors conclude that PLZF functions in enhancer activation and that the latent enhancer within its locus regulates the expression of *NNMT* during osteogenic differentiation. The findings are of interest, but there are several concerns with respect to the strength of some of the main conclusions.

Essential revisions:

1) The general pattern of PLZF binding is probably sound based on the absence of peaks in PLZF knockdown cells but the ChIP-seq experiments do not meet current standards for reproducibility. ChIPs from three independent experiments were pooled for sequencing. It is not possible from this experimental design to establish biological reproducibility of the peaks. This requires independent sequencing of ChIP products and application of statistical methods such as IDR. In addition, a table of general metrics for sequencing should be provided to help with data interpretation, including number of uniquely mapped reads, ChIP efficiency (fraction of reads in peaks) etc.

2) Specificity of siRNA knockdown. Although the methods section indicates that three different siRNAs were tested for PLZF, it is not evident that the three siRNAs generated the same result for PLZF or *NNMT*. This is important because of the potential for significant off-target effects of siRNAs. The striking effect of the *NNMT* siRNA on *PLZF* and *ALPL* expression during osteogenic differentiation should be clearly established by more than one siRNA.

3) Role of PLZF in enhancer activation. The ChIP-seq data indicates that PLZF primarily binds to a pre-existing enhancer landscape, with a significant faction of these regions showing an increase in H3K27ac. However, the cause-effect relationship between PLZF binding and gain in H3K27ac is not established. Many other factors are induced during osteogenic differentiation and could be the primary drivers of this change. A cause-effect relationship requires performing ChIP-seq for H3K27ac during osteogenic differentiation in cells in which PLZF has been knocked down.

4) Most importantly, there are no transcriptomic studies of the consequences of PLZF loss of function. Although it was shown that *Zbtb16* knockout leads to a skeletal defect (Barna et al., 2000), it is not necessarily an osteogenic defect. The only experiment evaluating effects of PLZF knockdown on gene expression is the Western blot for *NNMT* in Figure 6A, which does appear to show a difference at days 2 and 5 (but not day 10). However, there is no quantitative or statistical analysis of this finding and effects on mRNA levels are not provided. At this stage, there is no convincing evidence that PLZF plays an important role in regulating an osteogenic program of gene expression in this model system. This would require transcriptomic profiling in the PLZF knockdown cells.

5) The role of the enhancer EnP is not sufficiently dissected. While its activation is well supported by chromatin changes and by a retroviral enhancer assay, deletion of this enhancer should be performed, for example by CRISPR/Cas, to establish its physiological relevance. The 4C data in Figure 4 look rather unconvincing and are not statistically analyzed, and the impact of PLZF on *NNMT* seems weak.

6) There are many instances of figure panels lacking measures of statistical significance between experimental groups e.g., all of Figures 3, 5A, E, F, G, 6C, D, E, F.

[Editors’ note: the revision plan was rejected after consideration by the editors and reviewers, but the authors submitted for reconsideration. The decision letter issued after receiving the revision plan is shown below.]

Thank you for submitting your work entitled "PLZF targets developmental enhancers for activation during osteogenic lineage commitment of human mesenchymal stem cells" for consideration by *eLife*.

We have considered the response you provided with regard to your plan for a revision and we regret to inform you that your work will not be considered further for publication in *eLife*. One of the key conclusions from the Abstract of your manuscript was, 'Together, we show, PLZF functions in enhancer activation…' This conclusion was based on an association of PLZF binding with genomic regions that gain histone acetylation during the differentiation process. An essential revision was to perform ChIP-seq experiments in cells lacking PLZF to establish that H3K27ac was PLZF-dependent. The data provided in Figure 3 of the plan for revision indicates no change in the gain of H3K27ac at the enhancer EnP in the PLZF knockdown cells. This experiment argues against the main conclusion that PLZF targets developmental enhancers for activation during osteogenic lineage commitment of human mesenchymal stem cells, as stated in the title. Therefore, while you can address many of the other points raised as essential revisions, the central underlying mechanisms for how PLZF would alter programs of osteogenic differentiation are not established. Unfortunately, we think that the manuscript is no longer an appropriate candidate for *eLife*.

*Reviewer #1:*

In this article, the authors made a strong case that PLZF, encoded by *Zbtb16*, is involved in the osteogenic-differentiation of hMSC. PLZF binds to osteogenic enhancers. Furthermore, the upregulation of *Zbtb16* correlates with gain in H3K27 acetylation of osteogenic enhancers. Among these enhancers the authors demonstrated that an EnP enhancer occupied by PLZF within the PLZF locus loop to the neighboring gene *NNMT* and correlate with gain of *NNMT* expression during lineage commitment. Overall, the analysis of extensive genomic data is carefully done. However, there are several points need to be addressed to consolidate the conclusion that *Zbtb16* is a key osteogenic factor in MSC differentiation model.

1) Would knockout of *Zbtb16* by itself block osteogenic differentiation (instead of just correlate with osteogenic differentiation)? Although it was shown that *Zbtb16* knockout leads to skeletal defect (Barna et al., 2000), it is not necessarily an osteogenic defect (such as it could be chondrogenic defect).

2) Similarly, although the evidence that PLZF binds to active enhancers are strong, the evidence that the occupancy of PLZF leads to the active enhancer status is lacking. Would knockout/knockdown of *Zbtb16* obliterate H3K27ac binding signal in the PLZF bound enhancers? Or alternatively, would *Zbtb16* gain of function increase the H3K27ac signal across PLZF bound enhancers? What are the changes in transcriptome with PLZF loss/gain of function? What other factors bind to the same sets of enhancers PLZF bind in osteogenesis related context? (This can be done by leveraging published ChIP-seq data in the context of osteogenesis.)

3) The EnP enhancer is very interesting, could the author demonstrate that the EnP-GFP+ cells are in a more primed/committed state than GFP- cells? This can be done by FACS-GFP and RNA-seq. Given that the entire *PLZF-NNMT* locus is likely co-regulated by a cluster of enhancers (as can be inferred by H3K27ac signals across the regions), is there any functional data to show the importance of EnP enhancer in this context?

4) In the title it states that PLZF targets developmental enhancers. Is there any direct evidence to suggest that these enhancers are 'developmentally' involved? Such as cross reference with vista enhancer browser enhancers.

*Reviewer #2:*

Singh et al. report that PLZF encoded by *Zbtb16* is induced during differentiation of hMSCs into osteoblasts in vitro. In addition, they focus on an enhancer called EnP within the *Zbtb16* locus that is bound by the PLZF protein and loops to the promoter of the neighboring *NNMT* gene to regulate its expression. There are major gaps in this scenario.

The function of PLZF in human MSC is not established since the authors do not present loss or gain of function studies, only a ChIP-seq analysis that shows binding to some osteoblastic genes among thousands of genes. In addition, they mention that PLZF is similarly induced during adipocyte and chondrocyte differentiation, which casts doubt on its specificity for osteoblasts. In fact, there are no in vivo data or reference to support the contention that PLZF regulates osteoblastic differentiation in human. The mouse literature indicates a role of PLZF in skeletal patterning through regulation of HOX genes, not osteoblastic regulation. Finally, the PLZF ChIP-seq analysis is incomplete: the authors do not assess the presence of a DNA binding motif for PLZF, or compare their data with previous reports in human cells or in mouse NKT cells.

The role of the enhancer EnP is not sufficiently dissected. While its activation is well supported by chromatin changes and by a retroviral enhancer assay, deletion of this enhancer should be performed, for example by CRISPR/Cas, to establish its physiological relevance. The 4C data in Figure 4 look rather unconvincing and are not statistically analyzed, and the impact of PLZF on *NNMT* seems weak. Therefore, the author's scenario describing a looping of this enhancer to the neighboring NNMT gene promoter and the suggestion that this may regulate global functions of methylation that impact osteoblastic differentiation remain vague and hypothetical.

*Reviewer #3:*

The manuscript from Singh et al. reports on genomic roles of PLZF during lineage commitment of human mesenchymal stem cells and evidence for a latent enhancer within the *ZBTB16/PLZF* locus that is developmentally induced and regulates expression of the *NNMT* gene to influence cellular levels of SAM. The studies are of general interest with respect to understanding mechanisms of osteogenic differentiation. The authors demonstrate that PLZF is strongly induced at an early time point during osteogenic differentiation, coinciding with loss of the repressive H3K27me3 mark and gain of H3K27ac. ChIP-seq studies indicate that PLZF primarily binds to intra- and intergenic regions with features of enhancers, many of which exhibit increases in H3K267ac during differentiation and are near genes that are enriched for relevant functional annotations. Chromatin conformation capture (4C) experiments are performed indicating an interaction of the latent enhancer within the *ZBTB16* locus and the *NNMT* gene, which is not dependent on PLZF induction. Induction of the expression of *NNMT*, which catalyzes the conversion of nicotinomide to N1-methyl nicotinomide in a SAM-dependent manner, results in a modest but significant reduction of SAM levels. Knockdown of the *NNMT* using an siRNA results in a striking block of induction of *ZBTB16* and *ALPL* mRNA during osteogenesis. The authors conclude that PLZF functions in enhancer activation and that the latent enhancer within its locus regulates the expression of NNMT during osteogenic differentiation. The findings are of interest, but there are several concerns with respect to the strength of some of the main conclusions.

1) The general pattern of PLZF binding is probably sound based on the absence of peaks in PLZF knockdown cells but the ChIP-seq experiments do not meet current standards for reproducibility. ChIPs from three independent experiments were pooled for sequencing. It is not possible from this experimental design to establish biological reproducibility of the peaks. This requires independent sequencing of ChIP products and application of statistical methods such as IDR. In addition, a table of general metrics for sequencing should be provided to help with data interpretation, including number of uniquely mapped reads, ChIP efficiency (fraction of reads in peaks) etc.

2) Specificity of siRNA knockdown. Although the Materials and methods section indicates that three different siRNAs were tested for PLZF, it is not evident that the three siRNAs generated the same result for PLZF or NNMT. This is important because of the potential for significant off-target effects of siRNAs. The striking effect of the NNMT siRNA on PLZF and ALPL expression during osteogenic differentiation should be clearly established by more than one siRNA.

3) Role of PLZF in enhancer activation. The ChIP-seq data indicates that PLZF primarily binds to a pre-existing enhancer landscape, with a significant faction of these regions showing an increase in H3K27ac. However, the cause-effect relationship between PLZF binding and gain in H3K27ac is not established. Many other factors are induced during osteogenic differentiation and could be the primary drivers of this change. A cause-effect relationship requires performing ChIP-seq for H3K27ac during osteogenic differentiation in cells in which PLZF has been knocked down.

4) The studies provide limited evidence for a role of PLZF in osteogenic differentiation. Most importantly, there are no transcriptomic studies of the consequences of PLZF loss of function. The only experiment evaluating effects of PLZF knockdown on gene expression is the Western blot for NNMT in Figure 6A, which does appear to show a difference at days 2 and 5 (but not day 10). However, there is no quantitative or statistical analysis of this finding and effects on mRNA levels are not provided. At this stage, there is no convincing evidence that PLZF plays an important role in regulating an osteogenic program of gene expression in this model system. This would require transcriptomic profiling in the PLZF knockdown cells.

5) There are many instances of figure panels lacking measures of statistical significance between experimental groups i.e., all of Figure 3, 5A, E, F, G, 6C, D, E, F.

[Editors’ note: what now follows is the decision letter after the authors submitted for further consideration.]

Thank you for resubmitting your work entitled "PLZF targets developmental enhancers for activation during osteogenic differentiation of human mesenchymal stem cells" for further consideration at *eLife*. Your revised article has been favorably evaluated by Jessica Tyler as the Senior Editor, a Reviewing Editor, and 1 reviewer.

The manuscript has been improved but there are some remaining issues that need to be addressed before acceptance. The first relates to the fact that it was not possible to generate replicates for the ChIP-seq analysis of PLZF due to discontinuation of the original antibody. The reviewers concur that motif analysis of the existing data set yielded the correct motif and that locus-specific confirmation of PLZF binding was performed at key genomic locations using the original antibody for biological replicates. However, it is not clear whether the additional antibodies tested included an antibody recently demonstrated to work for PLZF ChIP-seq in human cells. Second, while new transcriptomic analysis of the consequences of loss of function of PLZF during osteogenic differentiation was performed, the analysis is difficult to follow and does not provide a coherent picture at this stage. To address these concerns, the following revisions/clarifications are required.

1) A commercial mAb AF2944 (R&D) against human PLZF was recently used for ChIP-seq in human KJ1a cells (Mao et al., 2016, Supplementary Figure 2). Did the authors attempt to use this antibody? If not, experiments to assess whether this antibody can be used to replicate the locus specific ChIP experiments should be performed. These experiments do not need to 'work' for acceptance of the manuscript, but a positive result would be important because they would indicate that a key reagent is available for replication and extension of this work.

2) The analysis of the RNA-seq data shown in Figure 3 is superficial and does clearly link loss of function of PLZF to the overall gene expression program illustrated in Figure 1. The analysis of this data set should be performed along the lines of that shown in Figure 1—figure supplement 1 in order to allow a clearer interpretation of the results.

References:

Mao AP, Constantinides MG, Mathew R, Zuo Z, Chen X, Weirauch MT, Bendelac A. Multiple layers of transcriptional regulation by PLZF in NKT-cell development. 2016. PNAS 113 (27) 7602-7607. DOI: 10.1073/pnas.1601504113

---

## [Author Response]

[Editors’ note: what follows is the authors’ plan to address the revisions.]

Essential revisions1) The general pattern of PLZF binding is probably sound based on the absence of peaks in PLZF knockdown cells but the ChIP-seq experiments do not meet current standards for reproducibility. ChIPs from three independent experiments were pooled for sequencing. It is not possible from this experimental design to establish biological reproducibility of the peaks. This requires independent sequencing of ChIP products and application of statistical methods such as IDR. In addition, a table of general metrics for sequencing should be provided to help with data interpretation, including number of uniquely mapped reads, ChIP efficiency (fraction of reads in peaks) etc.

We agree with the reviewers, that in order to perform statistical analysis to validate the reproducibility of PLZF ChIP-seq experiment, we have to perform the ChIP-seq using three independent biological replicates. We will repeat the experiment to fulfill this requirement and furthermore provide a table with the general metrics as the reviewers asked for.

Experimental set up and time-line:

Please note that primary human MSCs are slow growing and have a low nucleus to cytoplasmic ratio and big cell size (a confluent 15 cm plate contains app. 1.5x10^6 cells). Therefore, it will take more than one month to obtain sufficient cells to perform this ChIP-seq experiment.

Furthermore, for the sequencing of the ChIP-libraries we expect a turn-around time of approx. 3 weeks at our facility at BRIC. This will be followed by analyses that will take approx. 2-3 weeks.

2) Specificity of siRNA knockdown. Although the methods section indicates that three different siRNAs were tested for PLZF, it is not evident that the three siRNAs generated the same result for PLZF or NNMT. This is important because of the potential for significant off-target effects of siRNAs. The striking effect of the NNMT siRNA on PLZF and ALPL expression during osteogenic differentiation should be clearly established by more than one siRNA.

We do have data for three different siRNA oligos directed towards PLZF (labeled 55, 56 and 57) and NNMT (labeled 20, 21 and 22) that were analyzed at the same time as the siRNA used in the experiment presented in Figure 5E and Figure 6A in the submitted manuscript. The effect of the other two siRNAs for PLZF knockdown was similar to the siRNA used in Figure 6A in the manuscript (see Figure 3—figure supplement 1A, in which siRNA 57 represents the siRNA used in the manuscript Figure 6A).

The effect of the other two siRNAs against NNMT on PLZF and ALPL expression was quite similar to the experiment presented in Figure 5E of the submitted manuscript. Please see Figure 6—figure supplement 1.

3) Role of PLZF in enhancer activation. The ChIP-seq data indicates that PLZF primarily binds to a pre-existing enhancer landscape, with a significant faction of these regions showing an increase in H3K27ac. However, the cause-effect relationship between PLZF binding and gain in H3K27ac is not established. Many other factors are induced during osteogenic differentiation and could be the primary drivers of this change. A cause-effect relationship requires performing ChIP-seq for H3K27ac during osteogenic differentiation in cells in which PLZF has been knocked down.

We agree with reviewer that an increase in H3K27ac could be mediated by other factors than PLZF induced during osteogenic differentiation. This we already discussed in the discussion section of the submitted manuscript. However, in order to identify the cause-effect relationship between PLZF binding and H3K27ac, we have initially performed ChIP-QPCR experiment for H3K27ac before and after PLZF knockdown in hMSC. The data shown in Author response image 1 reveal that the increase in H3K27ac at the EnP enhancer is observed even in the absence of PLZF, suggesting the involvement of other factors that are induced and/or activated during osteogenic differentiation of hMSC.

**Author response image 1. respfig1:** ChIP-QPCR at the *ZBTB16* locus showing enrichment of H3K27ac histone mark in naive or osteogenic committed (2 days) hMSCs after PLZF knockdown. Data showing mean ± SD from triplicates of QPCR experiment. The promoter, TSS (Exon1) and EnP corresponds to primers A, B and E in Figure 1C in the submitted manuscript.

Moreover, as suggested by the reviewers, in order to reveal if there should be a gene specific effect we are going to perform the ChIP-seq for H3K27ac in PLZF knockdown cells during osteogenic differentiation and compare to control siRNA transfected cells. We will choose PLZF siRNA 57 for this experiment. The experiment will be performed in biological triplicates using antibodies for H3K27ac, PLZF, general H3 and IgG.

Experimental plan and time-line:

We will grow hMSCs in sufficient amount (4-5 weeks), followed by PLZF knockdown using PLZF siRNA-57 and control siRNA (Scrambled) transfected cells. Then osteogenic differentiation will be induced for 2 days and cells will be fixed for ChIP-seq for H3K27ac, PLZF, general H3 and IgG control. The experiment will be performed in biological triplicates in order to meet the standard for reproducibility.

Furthermore, for the sequencing of the ChIP-libraries we expect a turn-around time of approx. 3 weeks at our facility at BRIC. This will be followed by analyses that will take approx. 2-3 weeks. We will furthermore add a table of general metrics information regarding the ChIP-seq data.

4) Most importantly, there are no transcriptomic studies of the consequences of PLZF loss of function. Although it was shown that Zbtb16 knockout leads to a skeletal defect (Barna et al., 2000), it is not necessarily an osteogenic defect. The only experiment evaluating effects of PLZF knockdown on gene expression is the Western blot for NNMT in Figure 6A, which does appear to show a difference at days 2 and 5 (but not day 10). However, there is no quantitative or statistical analysis of this finding and effects on mRNA levels are not provided. At this stage, there is no convincing evidence that PLZF plays an important role in regulating an osteogenic program of gene expression in this model system. This would require transcriptomic profiling in the PLZF knockdown cells.

We had similar concern as the reviewers that it is important to understand the PLZF loss of function. Previously, we have done some initial experiments (data shown in Author response image 2), to study the effect of PLZF knockdown on osteogenic differentiation with ALPL and NNMT as readout. The 10 day time point shows a less pronounced effect and we believe this is related to PLZF being re-expressed due to a dilution effect or stability issue on the siRNA at this late time point.

**Author response image 2. respfig2:** A) PLZF knockdown using siRNA (oligo 57) in hMSCs measured by RT-QPCR at 2, 5 and 10 days of osteogenic induction. Relative mRNA levels were calculated after normalization to GAPDH. Data is an average from 3 independent biological replicates ± SD. B) ALPL, C) NNMT, mRNA expression analysed by RT-QPCR in hMSCs before and after osteogenic induction for 2, 5 and 10 days in siPLZF or control siRNA transfected cells. Relative mRNA levels are calculated after normalization to GAPDH.Data shows mean ± SD from triplicates of QPCRs from 3 independent experiments.

Additionally, we now have the whole transcriptome profiling before and after PLZF knockdown in hMSCs and the impact on expression of osteogenic induced genes. These data have been analysed and are shown in Author response image 3.

**Author response image 3. respfig3:** A) Transcriptome analyses by RNA-seq after siRNA (oligo 57) mediated PLZF knockdown followed by osteogenic induction, was performed in biological triplicates. The normalized expression was calculated using Genomatix software using their guidelines (Expression Analysis for RNASeq Data, GGATM) (Genomatix Software GmbH, Germany). As shown in Figure 5, the expression of genes induced during osteogenic differentiation were reduced in the absence of PLZF. Transcripts that were induced upon induction of osteogenic differentiation in control siRNA treated cells (log2 fold≥1, n=7,211) were sorted according to their expression levels in the control cells (left to right on the X-axis) and the expression levels measured in cells treated with PLZF siRNA was overlaid. The Y-axis indicates the mean normalized expression values from each group, control-siRNA (shown in blue) and PLZF-siRNA (shown in red). B) Fold change (fc) in expression (day 2 of osteogenic diff. /naïve hMSCs) calculated from normalized expression of selected osteogenic lineage specific genes in control siRNA and PLZF siRNA transfected hMSCs. The values are averages from triplicates of 3 independent biological experiments ± SD. ALPL, alkaline phosphatase; LEP, Leptin; SPON1, Spondin1; IGF2, Insulin Like Growth Factor 2; IGF2BP2, Insulin Like Growth Factor 2 mRNA binding Protein 2.

5) The role of the enhancer EnP is not sufficiently dissected. While its activation is well supported by chromatin changes and by a retroviral enhancer assay, deletion of this enhancer should be performed, for example by CRISPR/Cas, to establish its physiological relevance. The 4C data in Figure 4 look rather unconvincing and are not statistically analyzed, and the impact of PLZF on NNMT seems weak.

We thank reviewers for appreciating the data about enhancer activation. We agree that in order to establish the physiological relevance of EnP, the deletion of this enhancer by CRISPR/Cas9 would have been an excellent addition. However, the human mesenchymal stem cells that we have used throughout these studies can only be passaged until p11, after which they stop growing. Therefore, it will not be feasible to obtain a stable cell line of these hMSCs with a knockout of the EnP enhancer that will allow us to perform experiments to address its function. Furthermore, since EnP is an intragenic enhancer located within the PLZF locus, it is likely that the deletion would interfere with the genomic structure in such a way that it would directly influence on PLZF expression. Therefore, any observed phenotype would be difficult to interpret e.g. if it is due to the EnP deletion as such or due to a disruption of the locus structure that could influence e.g. splicing and exon usage.

We actually thought that the 4C-seq data were very convincing. However, we will do our best to find a solution to improve the presentation of the 4C-seq data to convince the reviewers, and furthermore include appropriate statistics.

6) There are many instances of figure panels lacking measures of statistical significance between experimental groups e.g., all of Figures 3, 5A, E, F, G, 6C, D, E, F.

We apologize for the missing statistical testing in the indicated experiments. This has been calculated and will be added as requested by the reviewers.

[Editors’ note: the author responses to the essential revisions after resubmitting their manuscript for consideration now follows.]

The manuscript from Singh et al. reports on genomic roles of PLZF during lineage commitment of human mesenchymal stem cells and evidence for a latent enhancer within the ZBTB16/PLZF locus that is developmentally induced and regulates expression of the NNMT gene to influence cellular levels of SAM. The studies are of general interest with respect to understanding mechanisms of osteogenic differentiation. The authors demonstrate that PLZF is strongly induced at an early time point during osteogenic differentiation, coinciding with loss of the repressive H3K27me3 mark and gain of H3K27ac. ChIP-seq studies indicate that PLZF primarily binds to intra and inter genic regions with features of enhancers, many of which exhibit increases in H3K267ac during differentiation and are near genes that are enriched for relevant functional annotations. Chromatin conformation capture (4C) experiments are performed indicating an interaction of the latent enhancer within the ZBTB16 locus and the NNMT gene, which is not dependent on PLZF induction. Induction of the expression of NNMT, which catalyzes the conversion of nicotinomide to N1-methyl nicotinomide in a SAM-dependent manner, results in a modest but significant reduction of SAM levels. Knockdown of the NNMT using an siRNA results in a striking block of induction of ZBTB16 and ALPL mRNA during osteogenesis. The authors conclude that PLZF functions in enhancer activation and that the latent enhancer within its locus regulates the expression of NNMT during osteogenic differentiation. The findings are of interest, but there are several concerns with respect to the strength of some of the main conclusions.

We highly appreciate that the reviewers and editor found our study of general interest for the understanding of mechanisms acting in the process of osteogenic differentiation of hMSCs. We are furthermore very thankful that we have got a chance to improve the manuscript for consideration for publication in *eLife*. What follows is a response to the essential revisions listed in the request for a revision plan.

Essential revisions:1) The general pattern of PLZF binding is probably sound based on the absence of peaks in PLZF knockdown cells but the ChIP-seq experiments do not meet current standards for reproducibility. ChIPs from three independent experiments were pooled for sequencing. It is not possible from this experimental design to establish biological reproducibility of the peaks. This requires independent sequencing of ChIP products and application of statistical methods such as IDR. In addition, a table of general metrics for sequencing should be provided to help with data interpretation, including number of uniquely mapped reads, ChIP efficiency (fraction of reads in peaks) etc.

Due to the very unfortunate situation that Santa Cruz discontinued the production of PLZF antibody that we used for ChIP (sc-22839) and that we were unsuccessful in obtaining an “old” aliquot by directly contacting the company we have not been able to repeat the PLZF ChIP. We have tested several other antibodies for PLZF in ChIP but without any success. Either they did not enrich chromatin or the antibodies were rather unspecific (also by testing in Western blotting). Therefore, we have further tested the PLZF peak data set that we included in the first submission of the manuscript to validate their relevance as discussed below.

As pointed out by the reviewers the PLZF binding pattern seems sound due to their disappearance in the PLZF KO situation. Based on the previous analyses and new validations, we strongly believe it is well controlled, which is based on the following:

1) Three independent ChIPs for PLZF from three independent experiment were pooled for sequencing.

2) The PLZF binding was confirmed by QPCR on target regions in completely independent experiments.

3) The loss of PLZF binding on several genomic targets was confirmed after PLZF knockdown. These data are included as a new Figure 2—figure supplement 1C-E in the revised manuscript.

4) Moreover, in order to further check the validity of PLZF peaks, we have performed a de novomotif analysis of PLZF peaks using the MEME suite at http://meme-suite.org and the MEME-ChIP tool set to search for new motifs in the discriminative mode. Discriminative de novomotif analysis of the DNA sequences within the PLZF peaks, compared to sequences from negative control regions, revealed a significant occurrence of a TACAGC motif (E = 1.1x10^-81^), which is highly similar to the TAC(T/A)GTA PLZF motif previously reported by Ivins et al. (2003), as shown in Figure 2—figure supplement 1A and B.

These data are included in subsection “PLZF binds to active chromatin regions in immature osteoblasts” and the Discussion section and presented in Figure 2—figure supplement 1.

We hope that the reviewers will agree with us, that this motif analysis supports the trueness of the identified PLZF binding sites in hMSCs that we previously presented.

2) Specificity of siRNA knockdown. Although the methods section indicates that three different siRNAs were tested for PLZF, it is not evident that the three siRNAs generated the same result for PLZF or NNMT. This is important because of the potential for significant off-target effects of siRNAs. The striking effect of the NNMT siRNA on PLZF and ALPL expression during osteogenic differentiation should be clearly established by more than one siRNA.

We have included the data for three different siRNA oligos directed towards PLZF (labeled #55, #56 and #57) presented in Figure 3—figure supplement 1 and the text. The effect of the other two siRNAs (#55, #56) on PLZF expression was similar to the siRNA #57 which is presented in the main Figures 3A and 7A in the revised manuscript.

For the *NNMT* knockdown three different siRNAs (labeled #20, #21 and #22) were tested and the data are presented in Figure 6—figure supplement 1, and the text. The other two siRNAs against NNMT affected *PLZF* and *ALPL* expression in a similar manner to the siRNA presented in Figure 6E-G in the revised manuscript. The siRNA labeled #22 was the one used in the main Figure 6E-G of the revised manuscript.

3) Role of PLZF in enhancer activation. The ChIP-seq data indicates that PLZF primarily binds to a pre-existing enhancer landscape, with a significant faction of these regions showing an increase in H3K27ac. However, the cause-effect relationship between PLZF binding and gain in H3K27ac is not established. Many other factors are induced during osteogenic differentiation and could be the primary drivers of this change. A cause-effect relationship requires performing ChIP-seq for H3K27ac during osteogenic differentiation in cells in which PLZF has been knocked down.

In order to identify the cause-effect relationship between PLZF binding and H3K27ac, we performed siRNA mediated PLZF knockdown in naïve hMSCs followed by induction of osteogenic differentiation followed by H3K27ac ChIP-seq as suggested by the reviewer.

The data revealed a highly reproducible increase in H3K27ac in pre-osteoblasts compared to naïve hMSCs (red in left panel), and this increase in H3K27ac seems to be PLZF dependent. The gain in H3K27ac at PLZF bound regions in pre-osteoblasts (red in left panel) were reduced when PLZF expression was down-regulated by siRNA. Examples of regions showing a gain of H3K27ac in a PLZF dependent manner are shown as ChIP-seq tracks in Figure 3C.

These data are included in the main Figure 3 and the text to describe these observations is included in subsection “PLZF affects histone H3K27ac at bound enhancers and an osteogenic gene expression signature”. The list of regions that gain H3K27ac in PLZF dependent manner are included as Table 7 in the revised manuscript.

It is of course very reasonable to assume that PLZF functions together with other transcription factors, that like PLZF are induced during osteogenic differentiation, and regulates genes important for osteogenic differentiation. This part is discussed in subsection “PLZF is required for the recruitment of Mediator and RNA PolII at the EnP enhancer”.

4) Most importantly, there are no transcriptomic studies of the consequences of PLZF loss of function. Although it was shown that Zbtb16 knockout leads to a skeletal defect (Barna et al., 2000), it is not necessarily an osteogenic defect. The only experiment evaluating effects of PLZF knockdown on gene expression is the Western blot for NNMT in Figure 6A, which does appear to show a difference at days 2 and 5 (but not day 10). However, there is no quantitative or statistical analysis of this finding and effects on mRNA levels are not provided. At this stage, there is no convincing evidence that PLZF plays an important role in regulating an osteogenic program of gene expression in this model system. This would require transcriptomic profiling in the PLZF knockdown cells.

We agree with the reviewers that it is important to understand the PLZF loss of function effect on transcription during osteogenic differentiation of hMSCs. We have now included novel data of the whole transcriptome profiling before and after PLZF knockdown in hMSC followed by osteogenic differentiation. We observed that PLZF knockdown significantly affected the osteogenic specific gene expression signature. The expression of genes that were normally induced in pre-osteoblasts is reduced in PLZF knockdown cells. These data are included in subsection “PLZF affects histone H3K27ac at bound enhancers and an osteogenic gene expression signature” and are shown in the main Figure 3D and E. Gene list is provided in Table 8.

5) The role of the enhancer EnP is not sufficiently dissected. While its activation is well supported by chromatin changes and by a retroviral enhancer assay, deletion of this enhancer should be performed, for example by CRISPR/Cas, to establish its physiological relevance. The 4C data in Figure 4 look rather unconvincing and are not statistically analyzed, and the impact of PLZF on NNMT seems weak.

We thank reviewers for appreciating the data about enhancer activation. We agree that in order to establish the physiological relevance of EnP, the deletion of this enhancer by CRISPR/Cas9 would have been an excellent addition. However, the human mesenchymal stem cells that we have used throughout these studies can only be passaged until p11, after which they stop growing and tend to differentiate. Therefore, it will not be feasible to obtain a stable cell line of these hMSCs with a knockout of the EnP enhancer that will allow us to perform experiments to address its function. Furthermore, since EnP is an intragenic enhancer located within the PLZF locus, it is likely that the deletion would interfere with the genomic structure in such a way that it would directly influence the expression of PLZF. Therefore, any observed phenotype would be difficult to interpret e.g. if it is due to the EnP deletion as such or due to a disruption of the locus structure that could influence e.g. splicing and exon usage.

We actually thought that the 4C-seq data were very convincing. However, we have re-analysed the data with statistical testing and presented it in the new Figure 5 in the revised manuscript.

6) There are many instances of figure panels lacking measures of statistical significance between experimental groups e.g., all of Figure 3, 5A, E, F, G, 6C, D, E, F.

We apologize for the missing statistical testing in the indicated experiments. This has been calculated by 2-way ANOVA with multiple comparison testings in all the experiments. The significant p values are indicated with asterisks in new figures and the exact values are mentioned in figure legends, (Figures 4B, 4C, 4F, 6A, 6B, 6D, 6E, 6F, 6G, 7A, 7B, 7C, 7D) in the revised manuscript.

[Editors' note: the author responses to the re-review follow.]

The manuscript has been improved but there are some remaining issues that need to be addressed before acceptance. The first relates to the fact that it was not possible to generate replicates for the ChIP-seq analysis of PLZF due to discontinuation of the original antibody. The reviewers concur that motif analysis of the existing data set yielded the correct motif and that locus-specific confirmation of PLZF binding was performed at key genomic locations using the original antibody for biological replicates. However, it is not clear whether the additional antibodies tested included an antibody recently demonstrated to work for PLZF ChIP-seq in human cells. Second, while new transcriptomic analysis of the consequences of loss of function of PLZF during osteogenic differentiation was performed, the analysis is difficult to follow and does not provide a coherent picture at this stage. To address these concerns, the following revisions/clarifications are required.1) A commercial mAb AF2944 (R&D) against human PLZF was recently used for ChIP-seq in human KJ1a cells (Mao et al., 2016, Supplementary Figure 2). Did the authors attempt to use this antibody? If not, experiments to assess whether this antibody can be used to replicate the locus specific ChIP experiments should be performed. These experiments do not need to 'work' for acceptance of the manuscript, but a positive result would be important because they would indicate that a key reagent is available for replication and extension of this work.

We have tested the mAb AF2944 from R&D systems in ChIP-QPCR on the same genomic loci as presented in the manuscript, Figure 2—figure supplement 1: the EnP enhancer, Col1A1 proximal, SMAD3A proximal. We observe that PLZF is enriched at these genomic regions after induction of osteogenic differentiation (2 days) as compared to naïve hMSCs, using the AF2944 antibody (please see Figure 2—figure supplement 1C-E). Therefore, we are very pleased to conclude, that this antibody can be considered as a replacement for the Santa Cruz PLZF antibody (sc-22839), which very unfortunately was discontinued by the company. This definitely will be helpful to reproduce and even extend our work in the future.

2) The analysis of the RNA-seq data shown in Figure 3 is superficial and does clearly link loss of function of PLZF to the overall gene expression program illustrated in Figure 1. The analysis of this data set should be performed along the lines of that shown in Figure 1—figure supplement 1 in order to allow a clearer interpretation of the results.

As per reviewer’s suggestion, we have re-analyzed the RNA-seq data from the PLZF knockdown experiment and is now presenting it in a more comprehensive way in revised Figure 3D and E, and Figure 3—figure supplement 1B and C. The heatmaps in Figure 3D, represents the RPKM-values of differentially regulated genes from RNA-seq of PLZF knockdown or control siRNA (in biological triplicates) in hMSCs. The vertical order of genes is similar to the clustered heatmap in Figure 1—figure supplement 1B. The RNA-seq data clearly indicate: i) a high degree of reproducibility in biological replicates, ii) pronounced impact of PLZF knockdown on gene regulation at an early stage (2 days) of osteogenic differentiation. Furthermore, in Figure 3E, we present the Beeswarm plots comparing the log2 fold differences between PLZF knockdown or negative control siRNA in hMSCs, harvested at day 0 and day 2 of osteogenic differentiation. The plot shows that PLZF knockdown significantly affected the osteogenic transcriptional program (p<0.0001). We also present the overlap by Venn diagram between differentially regulated genes in the absence or presence of PLZF during osteogenic induction and performed the Gene Ontology (GO) analyses of differentially regulated genes from each group (control siRNA or PLZF knockdown). All new analyses are now included in the main Figure 3 (revised) and Figure 3—figure supplement 1 (revised) and described in the main text, figure legends and Materials and methods section.